# Research on the impact and mechanism of local governments' land conveyance behavior on urban-rural integrated development—Empirical evidence from 281 prefecture-level cities in China

Long Zeng[1], Bin Peng[2]*, Jin Xie[3]

**1** School of Public Administration and Law, Hunan Agricultural University, Changsha, China, **2** School of Public Administration and Law, Hunan Agricultural University, Changsha, China, **3** School of Public Administration and Law, Hunan Agricultural University, Changsha, China

* q2660357334@163.com

## Abstract

The land sale practices of local governments significantly impact urban-rural integration development. This paper utilizes panel data from 281 prefecture-level and above cities in China, using local government land sale practices as a logical starting point. It examines the effects and mechanisms of these practices on urban-rural integration development from two dimensions: land price competition and land finance dependence, employing a bidirectional fixed effects model among other empirical tests. The findings indicate that land price competition has a significant positive effect on urban-rural integration development, showing an inverted U-shaped relationship. In contrast, land finance dependence hinders urban-rural integration development. The impact of local government land sale practices on urban-rural integration development also varies by region, administrative level, and different aspects of urban-rural integration. Furthermore, local government land sale practices mainly influence urban-rural integration development through labor transfer effects, public service provision effects, and industrial structure upgrading effects. Labor transfer and industrial structure upgrading play significant positive moderating roles in promoting urban-rural integration development through land price competition. Public service provision and industrial structure upgrading play significant positive moderating roles in mitigating the inhibitory effects of land finance dependence on urban-rural integration development.

## Introduction

The integrated development of urban and rural areas serves as a fundamental pathway for China's comprehensive rural revitalization and a crucial measure to dismantle the urban-rural duality and achieve common prosperity. As explicitly stated in the

**Data availability statement:** All relevant data are within the article and its Supporting Information files. The data used in this study are publicly available from the China Statistical Yearbook (https://www.stats.gov.cn/sj/ndsj/), the China Land and Resources Yearbook (https://www.las.ac.cn/front/book/detail?id=714b23fb0a1be7a2e4f-723f805a8a330), and the China Urban Statistical Yearbook (https://www.epsnet.com.cn/).

**Funding:** This research was made possible through the generous support of the National Natural Science Foundation of China (72303062; 72304095), the Humanities and Social Sciences Foundation of the Ministry of Education (22YJC790007), and the Natural Science Foundation of Hunan Province (2021JJ40263). Their financial assistance was instrumental in facilitating the successful completion of this research endeavor.

**Competing interests:** The authors have declared that no competing interests exist.

report of the 20th National Congress of the Communist Party of China, efforts should be made to promote urban-rural integration and coordinated regional development, facilitate the circulation of the national economy, and establish a new development dynamic between urban and rural areas. Consequently, how to achieve integrated urban-rural development has become a significant proposition in China's current phase of new development. From a theoretical perspective, research on urban-rural integrated development is rooted in theories of urban-rural relations, spatial economics, and institutional economics. Lewis's dual-sector model first revealed the economic structure of urban-rural segmentation in developing countries, highlighting the coexistence of traditional agricultural sectors and modern industrial sectors. The Todaro model further analyzed the economic motivations behind rural-urban migration, emphasizing the role of expected income differentials in labor mobility. Meanwhile, new economic geography explains, from a spatial perspective, how industrial agglomeration and dispersion shape urban-rural relations. These theories provide an essential analytical framework for China's pursuit of urban-rural integrated development.

Internationally, developed countries have cultivated relatively mature models for advancing urban-rural integration. The United States, for example, introduced the Smart Growth Program to establish metropolitan governance mechanisms, enhance infrastructure interconnectivity, and foster coordinated urban-rural planning. Germany implemented its "Urban-Rural Equalization" strategy, leveraging spatial planning laws and fiscal equalization to safeguard rural development rights. Japan successfully elevated rural industrial competitiveness through municipal mergers and the "One Village, One Product" initiative, while South Korea's New Village Movement systematically upgraded rural production and living standards through government-guided, community-driven participation. These cases collectively underscore that land institutional reform, equitable public service allocation, and industrial synergy constitute pivotal pathways to achieving urban-rural integration. From the process of promoting urbanization and rural revitalization in China, it is evident that adjusting the irrational allocation of resources can achieve balanced, shared, and integrated development between urban and rural areas [1]. As a spatial carrier and production factor of urban and rural regional systems and production activities, land's use and allocation changes impact the circulation of resources between urban and rural areas, playing a crucial role in transforming the urban-rural structure and driving economic growth [2].Given that the allocation of land elements in China is primarily government-led, local governments not only provide the foundational conditions for regional economic development by selling industrial land at low prices to attract foreign investment and industrial agglomeration but also generate financial resources for regional economic development by selling commercial and residential land at high prices [3]. This aids in promoting urban-rural integration development. However, the long-standing urban-biased development approach in China has led to issues such as land segmentation, separation of people and land, and urban-rural division. These issues can result in irrational land sale practices that hinder urban-rural integration development [4]. Thus, how do local government land sale practices impact urban-rural integration

development? Addressing this question not only clarifies the mechanisms by which local government land sale practices influence urban-rural integration development but also provides theoretical and practical bases for local governments to use land resource allocation to promote urban-rural integration development.

## Literature review

Under the urban-rural dual structure, cities typically offer more employment opportunities and higher wages, which incentivize the migration of labor and other production factors from rural to urban areas. However, with the evolving development environment between urban and rural regions, a reverse trend has emerged, where production factors, including labor, are increasingly flowing back from cities to rural areas. Urban-rural integrated development is widely regarded as a process that facilitates the rational flow and optimal allocation of production factors between urban and rural areas, thereby promoting economic integration and interactive growth. Existing literature has explored the conceptual framework and practical challenges of urban-rural integration from a theoretical perspective [5], while also identifying its driving mechanisms and pathways [6]. Empirical studies have further measured the level of urban-rural integration and examined its influencing factors from various dimensions [7], including economic growth [8], economic agglomeration [9], misallocation of urban-rural production factors [10], factor mobility [11], water quality improvement measures [12], new industrial cooperation models [13], and infrastructure development [14]. These studies often adopt analytical units at the provincial or prefecture-city level [15].

The academic community has deeply explored the relationship between land sales and urban-rural integration development. As a resource element and spatial carrier of urban and rural economic development, land has become a crucial force in the evolution of China's land system reform through market mechanisms [16]. Against the backdrop of a dual urban-rural land resource allocation system [17], urban-biased land acquisition methods [18], and government competition [19], land element allocation strategies and reforms have become important tools for local governments to develop regional economies [20–21].Overall, research on the relationship between land sales and urban-rural integration development can be divided into two main categories: Indirect Impact Studies: These studies examine how local government land sale practices indirectly influence urban-rural integration development by affecting related areas such as regional industrial structure upgrading [22], economic growth [23], urbanization development [24], urban-rural income gaps [25], Agricultural efficiency [26], sustainable development [27], and public services [28]. Direct Impact and Reform Path Studies: These studies explore the mechanisms [29–30], historical evolution, mechanism design [31], and institutional innovations [32] through which the Chinese land market impacts urban-rural integration development. On this basis, scholars further analyze the effects of land resource misallocation [11], land resource allocation efficiency [33], land use rates, and land transformation [34] on urban-rural integration development.

In summary, existing research has provided a significant theoretical and empirical foundation for this study. Existing literature predominantly focuses on the impacts of land markets, land resource allocation efficiency, and misallocation on urban-rural integration, while paying relatively little attention to the effects of local governments' land transfer behaviors on urban-rural integration. Furthermore, previous studies have not systematically conceptualized local governments' land transfer behaviors into the two dimensions of land price competition and land fiscal dependence to explore their impacts and heterogeneity concerning urban-rural integration. Additionally, the mechanisms through which local governments' land transfer behaviors influence urban-rural integration remain largely unexplored. The potential marginal contributions of this study are as follows: First, this paper is the first to systematically elucidate the impact mechanisms of local governments' land transfer behaviors on urban-rural integration, filling a gap in the existing literature. By constructing a unified theoretical framework, this study deeply examines how land transfer behaviors influence urban-rural integration through mechanisms such as labor migration, public service provision, and industrial structure upgrading. This provides a novel perspective for understanding the relationship between land transfer behaviors and urban-rural integration. Second, this study offers theoretical support and practical guidance for optimizing land transfer policies by local governments.

In particular, in the context of promoting urban-rural integration, the findings of this paper help local governments better understand the multidimensional impacts of land transfer behaviors, enabling them to formulate more scientific land transfer and fiscal policies. Furthermore, the policy recommendations provided in this study can help governments rationally adjust their land transfer practices to promote labor migration, enhance public service provision, and facilitate industrial structure upgrading, thereby fostering urban-rural integration, reducing the urban-rural gap, and achieving coordinated economic and social development.

## Theoretical basis and research hypotheses

As a critical foundation for regional economic and social development, land serves as a major fiscal revenue source for local governments under China's land concession system and the fiscal decentralization framework characterized by "centralized fiscal authority but decentralized administrative responsibilities. "With local governments holding absolute control over land concessions, land leasing and financing have become vital in supporting regional economic development and industrial policy formulation [35]. This model follows a dual institutional logic: On the one hand, Driven by political promotion incentives, local governments attract investment by leasing industrial land at low prices, creating employment opportunities and boosting economic growth [36]. For example, Jiangsu Province's 2021 Opinions on Further Promoting Industrial Land Efficiency emphasizes that industrial land is essential for sustaining real economic development—whether for upgrading traditional manufacturing or establishing new industrial bases. On the other hand, Under fiscal pressure from revenue-expenditure imbalances, local governments generate substantial income by leasing commercial/residential land at higher prices. This helps alleviate fiscal strain from decentralization while funding infrastructure projects [3]. For instance, Chaoyang City, Liaoning Province's 2021 budget report showed municipal government-managed fund revenues reaching ¥10.294 billion (103% of the target), a 131.7% increase—primarily due to rising real estate development activity and higher land concession revenues.The practices of local governments in land sales are further explained by two hypotheses:

Economic Performance Hypothesis: Local governments compete to offer low-cost industrial land to attract businesses, thus improving regional economic performance and achieving officials' promotion objectives [37].

Land Finance Hypothesis: Due to the fiscal gap created by the tax-sharing reform, where revenue collection is centralized, and expenditure responsibilities are decentralized, local governments face financial constraints [35]. Consequently, the prices and revenues from land sales significantly impact local governments' fiscal capacity, compelling them to rely on land finance for economic development. This reliance is manifested in the high prices charged for commercial and residential land sales to generate necessary fiscal revenue.

Based on the characteristics of local government land sale practices, land serves as a crucial means for local governments to attract investment and obtain fiscal revenue. The impact of these practices on urban-rural integration development is reflected in several ways. On the one hand, local governments adopting low or even zero-price policies for industrial land transfers can reduce the input costs for enterprises acquiring land, attracting investments from industrial and manufacturing enterprises while providing them with space for development. However, excessively low land transfer prices may lead to a "race to the bottom" among regions [38]. As the industrial land transfer price increases, it not only raises the entry threshold for enterprises—preventing the clustering of low-efficiency or mid- to low-end enterprises—but also incentivizes enterprises to upgrade machinery, adopt advanced production technologies, and enhance operational efficiency. These dynamics are conducive to promoting the flow of resources and factors between urban and rural areas, thereby advancing urban-rural integration.However, further increases in industrial land transfer prices may bring counter-productive effects. On the one hand, higher land prices raise the entry threshold for enterprises, causing some small and medium-sized enterprises (SMEs) to lose the ability to enter the market altogether. This reduction in market diversity and competitiveness weakens the overall dynamism of the economy. On the other hand, increased land prices impose higher costs on enterprises, discouraging investment. This weakens regional economic vitality, limits the inflow of factors such as capital, technology, and human resources, and restricts the mobility of regional resources. Ultimately, these constraints

hinder the optimal allocation of factors between urban and rural areas and obstruct the process of urban-rural integration. This reasoning provides a theoretical explanation for the observed inverted U-shaped relationship between land price competition and urban-rural integration [39], illustrating the dual impact of land price increases on resource allocation and economic development.

On the other hand, high prices for commercial and residential land sales generate significant fiscal revenue for local governments. This provides financial resources for regional infrastructure development and attracts more external investment, driving regional economic development. However, a higher reliance on land finance might lead local governments to adopt measures that segregate urban and rural land markets to protect their interests, hindering the flow of resources between urban and rural areas [33]. Additionally, rising land prices can lead to higher housing costs, increasing residents' living expenses and creating obstacles to their mobility between urban and rural areas. This could result in a rigid and homogenized industrial structure [22], exacerbating the irrational allocation of resources and making it difficult to break down urban-rural barriers, ultimately impeding urban-rural integration development. Based on the theoretical impact logic discussed above, we propose Hypothesis 1:

H1: Selling industrial land at low prices can effectively promote urban-rural integration development. However, excessively high industrial land prices will inhibit urban-rural integration development, resulting in an inverted U-shaped relationship between land price competition and urban-rural integration development. Additionally, reliance on land finance is detrimental to urban-rural integration development.

As previously mentioned, local government land sale practices often influence the flow of regional resources and elements. The transfer and aggregation of regional labor resources also provide a foundation for these land sale practices. Labor mobility is influenced by public services such as education and healthcare [40], and it can also be effectively attracted by industrial structure upgrading [41]. In this context, there is a significant interaction between labor resource mobility, public service provision, and industrial structure upgrading with local government land sale practices, which may, in turn, impact the effect of these practices on urban-rural integration development.Therefore, this study attempts to explore the mechanisms through which local government land sale practices influence urban-rural integration development by examining the interactive effects of labor transfer, public service provision, and industrial structure upgrading (Fig 1).

Labor Transfer Effects: The transfer of rural labor can meet the labor demand generated by urban industrial development, thereby promoting the flow of resources between urban and rural areas and impacting local government land sale practices on urban-rural integration development [42]. Specifically, in the case of selling industrial land at low prices, the transfer of rural labor increases the urban labor supply, which not only expands local government development and the supply of industrial land but also helps reduce labor input costs for enterprises and increases the demand for industrial land [43]. This strengthens the push of low-priced industrial land sales on urban-rural integration development.

On the other hand, in the case of selling commercial and residential land at high prices, rural labor transfer can drive urban population growth and increase demand for commercial and residential land. This has a dual effect: firstly, it can enhance local government land fiscal revenue, improving regional living conditions and public service facilities, thereby promoting urban-rural integration development. Secondly, it can lead to higher housing prices, making it more difficult for some incoming labor to afford homes, which may result in labor outflow. This counteracts the impact of labor transfer on land finance dependency affecting urban-rural integration development.Thus, we propose Hypothesis 2:

H2: The transfer of rural labor will enhance the positive moderating effect of land price competition on urban-rural integration development, but its impact on the effect of land finance dependency on urban-rural integration development is not significant.

Public Service Provision Effects: The enhancement of public service provision can alter the direction of resource flow between urban and rural areas [44], thus impacting how local government land sale practices influence urban-rural integration development. Specifically, regarding the low-price sale of industrial land, improved public service levels can enhance urban and rural infrastructure, creating a favorable environment for enterprises and

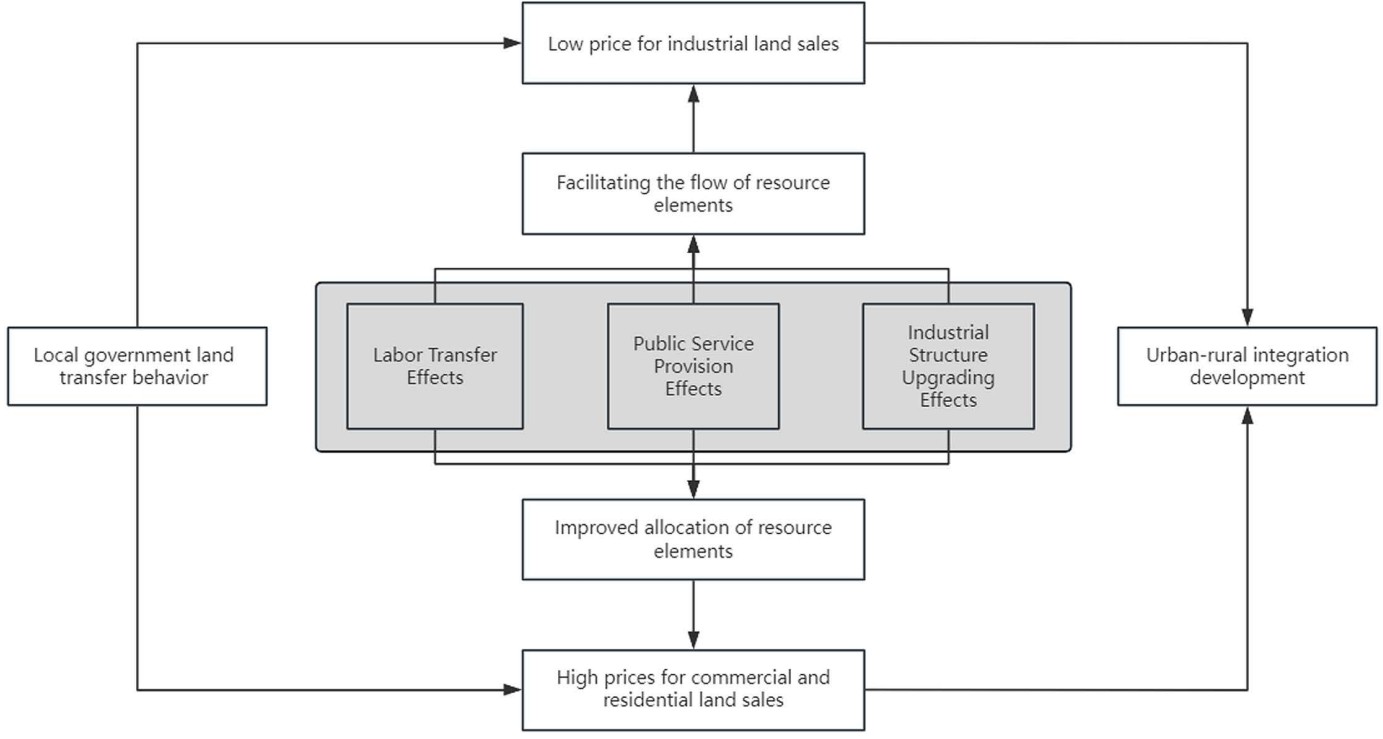

**Fig 1. Mechanism of local government land transfer behavior on urban-rural integration development.**

investors [45]. This can attract more external enterprises and increase the demand and scale of industrial land sales by local governments. Additionally, better public services can create a "voting with their feet" effect, encouraging the bidirectional flow of labor and capital, reducing the disparities in labor productivity and income between urban and rural areas [46], thereby strengthening the positive impact of low-price industrial land sales on urban-rural integration development.

In terms of the high-price sale of commercial and residential land, regions with higher levels of public services can attract more external population for employment and settlement, providing better social security and welfare. This enhances residents' sense of belonging and identity [47], effectively mitigating the pressure of high housing prices resulting from land finance dependency. Furthermore, improving public service levels can increase a region's attractiveness and competitiveness, driving economic growth and industrial diversification, which helps counteract the negative effects of land finance dependency on urban-rural integration development.Thus, we propose Hypothesis 3:

H3: The level of public service provision will positively moderate the impact of land price competition and land finance dependency on urban-rural integration development.

Industrial Structure Upgrading Effects: The process of industrial structure upgrading fosters the development of new industries, business models, and niche markets, promoting deeper industrial specialization and influencing the allocation of regional resources [48]. This, in turn, affects how local government land sale practices influence urban-rural integration development.Specifically, in the context of low-price industrial land sales, industrial structure upgrading can provide more employment opportunities for local labor and reduce the input costs for enterprises in terms of land and other production factors. This promotes the scale and agglomeration of regional industries, leveraging the synergistic effect of urban industries on rural industries, thereby enhancing the positive impact of low-price industrial land sales on urban-rural integration development.

Regarding the high-price sale of commercial and residential land, industrial structure upgrading signifies a shift towards diversified, knowledge-intensive, and service-oriented industries. Enterprises' cost budgets shift from traditional land acquisition towards green development and innovation research [49]. This can effectively mitigate issues related to high housing prices and rigid industrial structures caused by land finance dependency, improving the living environment and quality of life for residents [50]. Consequently, this weakens the negative impact of land finance dependency on urban-rural integration development.Thus, we propose Hypothesis 4:

H4: Industrial structure upgrading positively moderates both land price competition and land finance dependence affecting urban-rural integrated development.

## Materials and methods

### Model construction

**Establishment of baseline model.** Based on theoretical analysis and research hypotheses, this study employs a two-way fixed effects model to systematically examine the impact of local governments' land conveyance behavior on urban-rural integrated development. The model simultaneously controls for individual fixed effects and time fixed effects, effectively addressing estimation biases caused by time-invariant regional heterogeneity and region-invariant temporal factors. The specific model is specified as follows:

$$Degree_{it} = \beta_0 + \beta_1 Land_{it} + \delta control_{it} + \lambda_i + \phi_t + \varepsilon_{it} \tag{1}$$

In Equation (1), $Degree_{it}$ represents the level of urban-rural integration of city i in year t, $Land_{it}$ represents the behavior of local government land transfer, including the transfer of industrial land at a low price and the transfer of commercial and residential land at a high price, control denotes the set of control variables, $\lambda_i$, $\phi_t$ represent city and year fixed effects respectively, $\beta_0$ and $\beta_1$ are the coefficients to be estimated, $\varepsilon$ it is the error term used to absorb the influence of other confounding factors.

**Moderating effect model setting.** Furthermore, given the significant interactive relationships between local government land conveyance behavior and factors such as labor mobility, public service provision, and industrial structure upgrading—which may influence its impact on urban-rural integration—this study extends the baseline model by incorporating interaction terms. This approach allows for an examination of whether the effect of land conveyance is moderated by these channels. The extended moderated effects model is specified as follows:

$$Degree_{it} = \beta_0 + \beta_1 land_{it} + \beta_2 labor_{it} + \beta_3 land_{it} \times labor_{it} + \beta_4 control_{it} + \lambda_i + \phi_t + \varepsilon_{it} \tag{2}$$

$$Degree_{it} = \beta_0 + \beta_1 land_{it} + \beta_2 service_{it} + \beta_3 land_{it} \times service_{it} + \beta_4 control_{it} + \lambda_i + \phi_t + \varepsilon_{it} \tag{3}$$

$$Degree_{it} = \beta_0 + \beta_1 land_{it} + \beta_2 cyjg_{it} + \beta_3 land_{it} \times cyjg_{it} + \beta_4 control_{it} + \lambda_i + \phi_t + \varepsilon_{it} \tag{4}$$

Here, $labor_{it}$, $service_{it}$ and $cyjg_{it}$ represent labor migration, public service provision and industrial structure upgrading, $land_{it} \times labor_{it}$ denotes the interaction term between local government land granting behavior and labor transfer; $land_{it} \times service_{it}$ denotes the interaction term between local government land granting behavior and public service provision; $land_{it} \times cyjg_{it}$ denotes the interaction term between local government land granting behavior and industrial structure upgrading; $\beta_0$–$\beta_4$ are the coefficients to be estimated.

### Measurement and explanation of variables

**Dependent variable.** Urban-rural Integration Development (Degree), This study draws on the research of scholars such as Wang Ying [51] and Liao Zujun [10], focusing on the spatial-temporal evolution of urban-rural coordinated development in the Northeast China since 2003. It systematically constructs a comprehensive

evaluation index system for urban-rural integration development from five aspects: economic integration, social integration, population integration, spatial integration, and ecological integration (see Table 1). Based on the data from Table 1, the study employs the entropy method to measure the comprehensive index of urban-rural integration development.

Explanatory Variable: Local Government Land Transfer Behavior (Land), based on the analysis of local government land transfer behavior in the preceding sections and drawing from the approach of Chen Shuyun and Zeng Long [22], the core explanatory variables are defined as follows: ①Land Price Competition (lpc): Generally, local governments use negotiated transfer methods to attract external capital by offering industrial land at lower prices. Therefore, this study uses the unit price of land under negotiated transfers, defined as the ratio of revenue from negotiated transfers to the transferred area, as a proxy for land price competition. To mitigate the influence of inflation on land transfer prices, the price variable is adjusted using the Consumer Price Index (CPI) for each province where the cities are located, with 2003 as the base year. ②Land Fiscal Dependency (lfd): Higher-priced transfers of residential and commercial land enable local governments to generate more fiscal revenue for regional economic development. This has led to increased fiscal dependency on land revenues by local governments. Thus, the ratio of land transfer revenue to the budgeted fiscal revenue for each city is used as a proxy variable to measure the degree of land fiscal dependency. These variables are designed to capture the key aspects of local government land transfer strategies and their implications for economic development.

Moderating Variables: Including Labor Transfer Effects, Public Service Provision Effects, and Industrial Structure Upgrading Effects, as follows:

Labor Transfer Effects (labor): Drawing on Zeng Long and Yang Jiankun [52] approach, this paper defines the scale of labor transfer as:

$$labor_{i,t} = (L_{i,t} - L_{i,t-1}) - n_{i,t} \times L_{i,t-1}$$

(5)

Where $labor_{it}$ denotes the size of the labor transfer of city i in year t, $L_{it}$ denotes the total population at the end of the year of city i in year t, $n_{it}$ denotes the natural growth rate of the population of city i in year t. The calculation method indicates that labor migration is determined by the difference between the urban population at two points in time, subtracting the natural increase in the urban population. Specifically, the difference in stock between two time points represents the total population at the end of the current year minus the total population at the end of the previous year, reflecting the total

**Table 1. Comprehensive evaluation index system for urban-rural integration development.**

| Indicators | Level 1 | Level 2 | Properties | Weights |
|---|---|---|---|---|
| Degree | Economic | GDP per capita | + | 0.096 |
| | | Tertiary sector as a share of GDP | + | 0.016 |
| | | Ratio of disposable income per capita for urban and rural residents | | 0.004 |
| | | Value added of non-agricultural industries as a share of GDP | + | 0.012 |
| | Population | Urbanization rate | + | 0.024 |
| | | Urban population density | + | 0.078 |
| | Social | Share of education expenditure in total fiscal expenditure | + | 0.017 |
| | | Number of beds in hospitals and health centers | + | 0.057 |
| | Spatial | Built-up area as a proportion of administrative area | + | 0.271 |
| | | Postal and telecommunication operations per capita | + | 0.003 |
| | ecological | Green space per capita in parks | + | 0.183 |
| | | Sewage Discharge | | 0.235 |

population increase for that year. The natural increase in urban population is obtained by multiplying the total population at the end of the previous year by the natural growth rate.This calculation method separates the natural increase from the total population increase, explaining it as the "net migration" of urban population. This serves as a proxy variable for labor migration in this study.

Public Service Provision Effects (service): Drawing on Chen Shuyun and Zeng Long [53] approach, which utilizes the ratio of general fiscal expenditures to the total population at the end of the year.

Industrial Structure Upgrading Effects (cyjg): Drawing on Xu Ming, Jiang Yong [54]and Wang Wei [55] approach to construct the hierarchical index of industrial structure to represent the level of industrial structure upgrading, the specific measurement formula is as follows:

$$cyjg = \sum_{i=1}^{3} I_i \times i = I_1 + I_2 \times 2 + I_3 \times 3$$

(6)

Where Ii denotes the ratio of the industry's output to total output. Generally speaking, the index mainly reflects the upgrading relationship between the three industries, and the larger the index means the higher the level of upgrading the industrial structure of the region.

Control Variables: ①Degree of Openness (jck). The degree of openness of a region can impact urban-rural integration through industrial linkages and talent mobility effects. In this study, the degree of openness is measured by the ratio of total import and export volume to GDP. ②Level of Human Capital (rlzb). Urban-rural integration is significantly influenced by regional human capital factors. Human capital promotes integration by fostering creativity and enhancing the region's capacity for technological innovation. This study uses the ratio of the number of higher education students enrolled to the total year-end population to represent the level of human capital. ③Degree of Government Intervention (zfgy). Under the context of the urban-rural dual structure, local governments' fiscal expenditures often exhibit an urban bias, which affects urban-rural integration. This study measures the degree of government intervention using the ratio of local government fiscal expenditures to GDP. ④Urbanization Level (czh). Urbanization influences urban-rural integration through effects such as population migration and factor mobility. The urbanization level is measured in this study by the urbanization rate of each region. ⑤Infrastructure Level (dlmj). Improved infrastructure facilitates population inflows and industrial development in a region, thereby impacting urban-rural integration. This study measures the infrastructure level using the per capita road area of the region.

## Data sources and descriptive statistics

This study employs panel data of Chinese prefecture-level and above cities from 2003 to 2017. The initial sample included all prefecture-level cities nationwide. However, to ensure research consistency and comparability, we excluded certain cities (e.g., Lhasa, Zhongwei, Chaohu, and Bijie) due to data deficiencies or administrative division adjustments, ultimately retaining a balanced panel of 281 cities. Notably, the study period concludes in 2017 as the primary data source - China Land and Resources Yearbook - ceased publication thereafter, making subsequent land transaction price data unavailable. Data primarily sourced from "China Statistical Yearbook," "China Land and Resources Yearbook," and "China Urban Statistical Yearbook," with occasional missing data supplemented from provincial statistical yearbooks, city statistical yearbooks, China Urban-Rural Construction Database, and China Regional Economic Database, using interpolation where necessary. In addition, to minimize heteroscedasticity and reduce the influence of outliers on the model, this study introduces certain variables into the equations in logarithmic form based on their magnitudes. For instance, variables such as land price competition, urbanization level, and infrastructure level are log-transformed to improve the model's fit and robustness. Table 2 provides descriptive statistics for the main variables.

**Table 2. Descriptive statistics of variables.**

| | (1) | (2) | (3) | (4) | (5) | (6) |
|---|---|---|---|---|---|---|
| Variant(unit) | Name | Obs | Mean | Std. Dev. | Min | Max |
| Urban and rural integration and development | Degree | 4215 | 0.0872 | 0.0679 | 0.0196 | 0.7218 |
| Land price competition | lpc | 4215 | 348.0809 | 1,394.1940 | 0.0003 | 83,041 |
| Land finance dependence | lfd | 4215 | 0.5743 | 0.4144 | 0.0013 | 3.3421 |
| Labor force transfer | labor | 4215 | 0.6876 | 14.0903 | −267.7 | 315.0439 |
| Public service provision(yuan/person) | service | 4215 | 0.5383 | 0.6049 | 0.0312 | 10.9378 |
| Upgrading of industrial structure | cyjg | 4215 | 2.2312 | 0.1422 | 1.8223 | 2.8013 |
| Degree of openness | jck | 4215 | 0.2221 | 0.4337 | 1.00e-08 | 8.1339 |
| Human capital level | rlzb | 4215 | 0.0148 | 0.0210 | 1.00e-15 | 0.1311 |
| Government intervention level | zfgy | 4215 | 0.1632 | 0.1100 | 0.0028 | 2.2667 |
| Urbanization level | czh | 4215 | 50.1411 | 16, 7467 | 8.0506 | 100 |
| Infrastructure level(m²) | dlmj | 4215 | 13.9676 | 6.5684 | 0.3900 | 60.0700 |

## Results and discussion

### Panel unit root and cointegration tests

To avoid spurious regression, it is first necessary to perform unit root tests on the panel data to determine its stationarity. This study employs both the IPS test and LLC test, which are commonly used for homogeneous unit root testing, to examine the stationarity of each variable in the panel data. If both tests reject the null hypothesis of a unit root, it indicates that the variable is stationary. Conversely, if the null hypothesis of a unit root is accepted, it implies that the variable is non-stationary.

The test results, as shown in Table 3, indicate that for all variables, the first-order differenced results reject the null hypothesis of "the existence of a unit root" at the 1% significance level. This confirms that all variables are stationary, thereby ruling out the possibility of spurious regression. However, due to the potential instability of panel data, directly applying the ordinary least squares (OLS) method may still result in spurious regression. Therefore, a panel cointegration test is performed next to analyze whether the relevant variables exhibit cointegration relationships.

Based on the unit root test results, this study further conducts a cointegration test to verify the equilibrium relationship among the variables. Specifically, the Pedroni test, Westerlund test, and Kao test are employed. As shown in Table 4, the p-values of all test statistics are less than 0.01. Therefore, the null hypothesis of no cointegration is rejected at the 1% significance level, indicating the existence of a stable long-term cointegration relationship among the variables. This provides a solid foundation for conducting subsequent regression analysis.

### Benchmark regression results

Table 5 presents the baseline regression results on the impact of local government land transfer behaviors on urban-rural integration development. The regression results indicate that the estimated coefficients and significance levels of the core explanatory variables, whether examining land price competition or land fiscal dependence, align with expected outcomes. Specifically, regressions (1)-(3) demonstrate that the core explanatory variable of land price competition has a significant positive effect on urban-rural integration development. Conversely, an increase in land fiscal dependence shows a significant negative impact on urban-rural integration development. This suggests that lower-priced industrial land transfers can reduce enterprise input costs, promote industrial agglomeration and scale operation, and stimulate urban-to-rural spillover effects, thereby advancing urban-rural integration development. On the other hand, high-priced commercial and residential land transfers lead to issues such as high housing prices and industrial structure homogenization, which inhibit

**Table 3. Results of unit root test for variables.**

| Variables | | IPS | LLC | Conclusion |
|-----------|-----------|-----------|-----------|-----------|
| D(Degree) | Statistic | −25.0291*** | −16.7626*** | stable |
| | P | 0.0000 | 0.0000 | |
| D(lpc) | Statistic | −34.3859*** | −32.7703*** | stable |
| | P | 0.0000 | 0.0000 | |
| D(lfd) | Statistic | −34.1674*** | −36.0627*** | stable |
| | P | 0.0000 | 0.0000 | |
| D(jck) | Statistic | −21.7426*** | −5.0324*** | stable |
| | P | 0.0000 | 0.0000 | |
| D(rlzb) | Statistic | −2.7442*** | −63.9612*** | stable |
| | P | 0.0030 | 0.0000 | |
| D(zfgy) | Statistic | −31.7085*** | −28.4874*** | stable |
| | P | 0.0000 | 0.0000 | |
| D(czh) | Statistic | −23.0573*** | −1.5e+02*** | stable |
| | P | 0.0000 | 0.0000 | |
| D(dlmj) | Statistic | −25.3438*** | −31.6995*** | stable |
| | P | 0.0000 | 0.0000 | |

Note: D denotes first-order differencing, and *, **, *** represent significance levels of 10%, 5%, and 1%, respectively, at which the null hypothesis is rejected.

**Table 4. Cointegration test results.**

| Inspection type | Inspection Statistics | Statistics | P |
|-----------------|----------------------|------------|-----|
| Pedroni | Modified Phillips-Perron t | 30.0642 | 0.0000 |
| | Phillips-Perron t | −21.0833 | 0.0000 |
| | Augmented Dickey-Fuller t | −16.4603 | 0.0000 |
| Westerlund | Variance ratio | 22.7037 | 0.0000 |
| Kao | Modified Dickey-Fuller t | 8.5473 | 0.0000 |
| | Dickey-Fuller t | 8.0558 | 0.0000 |
| | Augmented Dickey-Fuller t | 13.9056 | 0.0000 |
| | Unadjusted modified Dickey-Fuller t | 2.6583 | 0.0039 |
| | Unadjusted Dickey-Fuller t | 1.6721 | 0.0472 |

the flow of urban-rural resource elements, thereby exerting a negative impact on urban-rural integration development. Furthermore, as shown in Regression (4), the significantly negative coefficient of the quadratic term for land price competition confirms an inverted U-shaped relationship between land price competition and urban-rural integrated development. This finding theoretically aligns with the threshold effect of industrial land transfers documented by Yang et al. [36], while providing empirical support for our earlier Hypothesis H1.

## Heterogeneity analysis

**Regression results and analysis by region.** Due to the current economic disparities among regions in China and the varied levels and modes of economic development across cities, the impact of local government land transfer behaviors on urban-rural integration development may vary. Following national classification standards and practices in relevant literature, this study divides the sample into Eastern, Central, and Western regions for regression analysis, as shown in Table 6.

**Table 5. Benchmark regression results.**

|  | (1) | (2) | (3) | (4) |
|---|---|---|---|---|
|  | Degree | Degree | Degree | Degree |
| lpc | 0.0009*** |  | 0.0009*** | 0.0015*** |
|  | (0.0003) |  | (0.0003) | (0.0005) |
| lfd |  | −0.0014** | −0.0016** |  |
|  |  | (0.0007) | (0.0007) |  |
| (lpc)2 |  |  |  | −0.0016* |
|  |  |  |  | (0.0009) |
| jck | −0.0013*** | −0.0013*** | −0.0013*** | −0.0013*** |
|  | (0.0003) | (0.0003) | (0.0003) | (0.0003) |
| rlzb | −0.0001 | −0.0001 | −0.0001 | −0.0001 |
|  | (0.0001) | (0.0001) | (0.0001) | (0.0001) |
| zfgy | −0.0030*** | −0.0031*** | −0.0031*** | −0.0031*** |
|  | (0.0008) | (0.0008) | (0.0008) | (0.0008) |
| czh | −0.0052*** | −0.0056*** | −0.0052*** | −0.0053*** |
|  | (0.0019) | (0.0019) | (0.0019) | (0.0019) |
| dlmj | −0.0077*** | −0.0075*** | −0.0076*** | −0.0077*** |
|  | (0.0008) | (0.0008) | (0.0008) | (0.0008) |
| _cons | 0.0879*** | 0.0938*** | 0.0884*** | 0.0899*** |
|  | (0.0078) | (0.0076) | (0.0078) | (0.0079) |
| N | 4215 | 4215 | 4215 | 4215 |
| r2 | 0.5624 | 0.5616 | 0.5630 | 0.5627 |
| Year fixed | Yes | Yes | Yes | Yes |
| City fixed | Yes | Yes | Yes | Yes |

Note:***, **, and * indicate significance levels at 1%, 5%, and 10%, respectively. The numbers in parentheses represent standard errors. The same applies to the following tables.

From the regression results in Table 6, it is evident that there is significant regional heterogeneity in the impact of local government land transfer behaviors on urban-rural integration development. The empirical results reveal significant regional heterogeneity: the coefficient of land price competition in western China is significantly positive at the 1% level, whereas the estimated coefficients for eastern and central regions are statistically insignificant. These findings indicate that land price competition in western China has a significant positive effect on urban-rural integrated development, while no significant impact is observed in eastern and central regions. The potential reasons for this are that Western China possesses abundant natural resources such as energy, minerals, and water, providing a crucial foundation for regional industrial development. Additionally, lower labor costs in the Western region make it more attractive to labor-intensive industries, effectively attracting related enterprises and capital aggregation, thereby promoting urban-rural integration development significantly through competitive land prices. In China's eastern and central regions, industrial development occurred relatively early, and economic development has gradually transitioned from traditional heavy industries to high-tech and service industries [56]. The relatively high labor costs in these regions have made low-priced industrial land transfers less effective in attracting enterprises, thereby diminishing their impact on urban-rural integration.

The regression results on fiscal reliance on land revenue indicate that the estimated coefficient for western China is significantly negative at the 1% level, while that of eastern China is statistically insignificant and central China shows a significantly positive coefficient. This suggests that fiscal dependence on land sales in western regions exerts a significantly negative effect on urban-rural integrated development, whereas it has a positive impact in central regions and

Table 6. Regression results by region.

| | (1) Eastern part | | (2) Central Region | | (3) Western region | |
|---|---|---|---|---|---|---|
| lpc | 0.0008 | | −0.0001 | | 0.0012*** | |
| | (0.0006) | | (0.0003) | | (0.0003) | |
| lfd | | −0.0019 | | 0.0016** | | −0.0037*** |
| | | (0.0013) | | (0.0008) | | (0.0010) |
| jck | −0.0083*** | −0.0085*** | 0.0001 | 0.0001 | −0.0001 | −0.0001 |
| | (0.0014) | (0.0014) | (0.0004) | (0.0004) | (0.0003) | (0.0003) |
| rlzb | −0.00007 | −0.00007 | 0.00001 | 0.00002 | 0.00002 | 0.00003 |
| | (0.00022) | (0.00022) | (0.00008) | (0.00008) | (0.00005) | (0.00005) |
| zfgy | −0.0029 | −0.0029 | −0.0009 | −0.0010 | −0.0027*** | −0.0030*** |
| | (0.0018) | (0.0018) | (0.0008) | (0.0008) | (0.0010) | (0.0010) |
| czh | −0.0081 | −0.0089* | 0.0163*** | 0.0162*** | −0.0208*** | −0.0219*** |
| | (0.0052) | (0.0052) | (0.0018) | (0.0018) | (0.0026) | (0.0026) |
| dlmj | −0.0159*** | −0.0158*** | −0.0044*** | −0.0045*** | 0.0002 | 0.0005 |
| | (0.0018) | (0.0018) | (0.0010) | (0.0010) | (0.0010) | (0.0010) |
| _cons | 0.1326*** | 0.1407*** | 0.0093 | 0.0086 | 0.1108*** | 0.1199*** |
| | (0.0213) | (0.0210) | (0.0076) | (0.0074) | (0.0100) | (0.0097) |
| N | 1500 | 1500 | 1485 | 1485 | 1230 | 1230 |
| r2 | 0.6013 | 0.6013 | 0.6638 | 0.6648 | 0.6484 | 0.6490 |
| Year fixed | Yes | Yes | Yes | Yes | Yes | Yes |
| City fixed | Yes | Yes | Yes | Yes | Yes | Yes |

no significant effect in eastern regions. A plausible explanation lies in regional economic disparities: In western China, underdeveloped economies with lower household incomes and consumption levels discourage diversified corporate investments [57]. This exacerbates structural rigidity and industrial homogenization caused by local governments' reliance on high-priced commercial/residential land sales, hindering urban-rural integration. In central China, rapid economic growth aligns with the State Council's 2021 Guidelines on Promoting High-Quality Development in the New Era, which emphasize building an advanced manufacturing-led modern industrial system. Local governments thus incentivize infrastructure investments using land-sale revenues to facilitate industrial relocation from eastern regions. Moreover, the issues related to high housing prices caused by land fiscal dependence are relatively less prominent in Central China, which further promotes the flow of resource elements such as labor between urban and rural areas, thus contributing significantly to urban-rural integration development. In Eastern China, the more developed economy allows local governments to have diverse and stable sources of fiscal revenue, reducing their dependence on land fiscal revenues. Additionally, Eastern China benefits from a diversified and reasonable industrial structure, excellent infrastructure conditions, and a robust public service system, which effectively mitigate negative effects such as rising housing prices and industrial structure rigidity caused by land fiscal dependence, resulting in its less significant negative impact on urban-rural integration development.

**Regression results and analysis by administrative level.** Considering the significant role of administrative hierarchy in resource allocation during the formation and development of cities, this study follows the approach of Nian Meng and Wang Yao [58] by categorizing directly-administered municipalities and provincial capitals as high administrative levels, while treating other cities as low administrative levels for analysis. As shown in Table 7, the regression results indicate that both the positive impact of land price competition on urban-rural integration development and the negative impact of land fiscal dependence are significant only in low administrative level cities.

**Table 7. Regression results by administrative level.**

| | (1) Low administrative level | | (2) High administrative level | |
|---|---|---|---|---|
| lpc | 0.0009*** | | 0.0006 | |
| | (0.0003) | | (0.0008) | |
| lfd | | −0.0012** | | −0.0025 |
| | | (0.0006) | | (0.0025) |
| jck | −0.0012*** | −0.0012*** | 0.0034 | 0.0035* |
| | (0.0003) | (0.0003) | (0.0021) | (0.0021) |
| rlzb | 0.00002 | 0.00002 | −0.0039 | −0.0038 |
| | (0.00005) | (0.00005) | (0.0025) | (0.0025) |
| zfgy | −0.0015** | −0.0015** | −0.0094*** | −0.0094*** |
| | (0.0007) | (0.0007) | (0.0028) | (0.0028) |
| czh | 0.0043** | 0.0040** | 0.0191 | 0.0169 |
| | (0.0017) | (0.0017) | (0.0127) | (0.0128) |
| dlmj | −0.0034*** | −0.0033*** | −0.0244*** | −0.0244*** |
| | (0.0007) | (0.0007) | (0.0033) | (0.0033) |
| _cons | 0.0411*** | 0.0464*** | 0.0737 | 0.0871 |
| | (0.0068) | (0.0066) | (0.0549) | (0.0553) |
| N | 3765 | 3765 | 450 | 450 |
| r2 | 0.5667 | 0.5658 | 0.8174 | 0.8176 |
| Year fixed | Yes | Yes | Yes | Yes |
| City fixed | Yes | Yes | Yes | Yes |

The possible reasons for these findings are as follows: In terms of low-priced industrial land transfers, high administrative level cities possess abundant governmental resources, innovation capabilities, and stronger economic prowess. This diminishes the incentive for local governments to utilize low-priced industrial land transfers to attract talent and high-tech enterprises, thereby resulting in a less significant impact on urban-rural integration development. Conversely, in low administrative level cities, lacking administrative power to facilitate resource aggregation and facing relatively lower income and consumption levels, local governments are more motivated to create more job opportunities through low-priced industrial land transfers. This effort promotes the flow of urban-rural resource elements and thus significantly contributes to urban-rural integration development.

Regarding high-priced commercial and residential land transfers, high administrative level cities leverage their strong fiscal capabilities and resource redistribution abilities to effectively radiate urban influence into rural areas. This helps alleviate issues such as industrial structure rigidity and high housing prices caused by land fiscal dependence, resulting in a non-significant impact of land fiscal dependence on urban-rural integration development. In contrast, low administrative level cities with weaker administrative capacity and limited fiscal revenue sources tend to overly rely on land fiscal policies for economic development. This dependency significantly hampers urban-rural integration development.

In summary, the impact of local government land transfer behaviors on urban-rural integration development exhibits significant regional heterogeneity influenced by administrative hierarchy in China. High administrative level cities show different patterns of impact compared to low administrative level cities, reflecting varying levels of resource allocation capability and economic conditions across administrative tiers.

**Regression results and analysis by urban-rural integration dimension.** To further analyze the impact of local government land transfer behaviors on urban-rural integration development across different dimensions—economic integration, social integration, population integration, spatial integration, and ecological integration—the regression results are presented in Table 8.From the regression results in Table 6, it is evident that:

**Table 8. Regression results by urban-rural integration dimension.**

| | (1) Economic integration | | (2) Population integration | | (3) Social integration |
|---|---|---|---|---|---|
| lpc | 0.0007 | | 0.0012*** | | 0.0011** |
| | (0.0004) | | (0.0002) | | (0.0005) |
| lfd | | −0.0048*** | | 0.0001 | |
| | | (0.0011) | | (0.0006) | |
| jck | −0.0020*** | −0.0021*** | −0.0003 | −0.0003 | 0.0005 |
| | (0.0006) | (0.0006) | (0.0003) | (0.0003) | (0.0006) |
| rlzb | 0.0002** | 0.0002** | −0.0002*** | −0.0002*** | −0.0007*** |
| | (0.0001) | (0.0001) | (0.0001) | (0.0001) | (0.0001) |
| zfgy | −0.0038*** | −0.0039*** | −0.0013* | −0.0013* | −0.0028** |
| | (0.0012) | (0.0012) | (0.0007) | (0.0007) | (0.0013) |
| czh | −0.0181*** | −0.0184*** | 0.0920*** | 0.0914*** | 0.0086*** |
| | (0.0032) | (0.0032) | (0.0017) | (0.0017) | (0.0033) |
| dlmj | −0.0037*** | −0.0035** | −0.0065*** | −0.0063*** | −0.0073*** |
| | (0.0014) | (0.0014) | (0.0007) | (0.0008) | (0.0014) |
| _cons | 0.2066*** | 0.2127*** | −0.1291*** | −0.1219*** | 0.0197 |
| | (0.0127) | (0.0125) | (0.0070) | (0.0069) | (0.0133) |
| N | 4215 | 4215 | 4215 | 4215 | 4215 |
| r2 | 0.7477 | 0.7488 | 0.7489 | 0.7473 | 0.5092 |
| Year fixed | Yes | Yes | Yes | Yes | Yes |
| City fixed | Yes | Yes | Yes | Yes | Yes |

| | (3) Social integration | (4) Spatial integration | | (5) Ecological integration | |
|---|---|---|---|---|---|
| lpc | | 0.0012*** | | 0.0006* | |
| | | (0.0004) | | (0.0004) | |
| lfd | −0.0001 | | −0.0009 | | −0.0013 |
| | (0.0012) | | (0.0009) | | (0.0009) |
| jck | 0.0005 | −0.0013*** | −0.0013*** | −0.0020*** | −0.0020*** |
| | (0.0006) | (0.0005) | (0.0005) | (0.0005) | (0.0005) |
| rlzb | −0.0007*** | −0.0002** | −0.0002** | 0.0001 | 0.0001 |
| | (0.0001) | (0.0001) | (0.0001) | (0.0001) | (0.0001) |
| zfgy | −0.0028** | −0.0036*** | −0.0037*** | −0.0032*** | −0.0032*** |
| | (0.0013) | (0.0010) | (0.0010) | (0.0010) | (0.0010) |
| czh | 0.0081** | −0.0118*** | −0.0123*** | −0.0245*** | −0.0248*** |
| | (0.0033) | (0.0027) | (0.0027) | (0.0026) | (0.0026) |
| dlmj | −0.0071*** | −0.0088*** | −0.0086*** | −0.0084*** | −0.0082*** |
| | (0.0014) | (0.0012) | (0.0012) | (0.0011) | (0.0011) |
| _cons | 0.0260** | 0.0750*** | 0.0823*** | 0.1296*** | 0.1339*** |
| | (0.0131) | (0.0107) | (0.0105) | (0.0104) | (0.0101) |
| N | 4215 | 4215 | 4215 | 4215 | 4215 |
| r2 | 0.5085 | 0.1582 | 0.1563 | 0.2784 | 0.2783 |
| Year fixed | Yes | Yes | Yes | Yes | Yes |
| City fixed | Yes | Yes | Yes | Yes | Yes |

Note:Due to table length limitations, Table 8 divides the social integration dimension into two parts: the upper section presents the regression results of land price competition on urban-rural social integration, while the lower section displays the regression results of land fiscal dependence on urban-rural social integration.

On one hand, land price competition shows non-significant effects on the economic integration dimension of urban-rural integration development. This might be due to the fact that as the agreed land transfer prices increase, although they discourage the aggregation of inefficient or mid-to-low-end enterprises and encourage enterprises to upgrade machinery and change production technologies, they also raise entry barriers and input costs for businesses. Consequently, this non-significant impact on economic integration development characterized by economic growth and industrial development. On the other hand, land fiscal dependence exhibits a significant negative impact only on the economic integration dimension of urban-rural integration. Therefore, for local governments, it is crucial to address the negative impact of land fiscal dependence on regional economic integration development by reducing excessive reliance on land fiscal policies. Shifting land fiscal expenditures from economic centers towards public services such as social security and public services can enhance the attractiveness of the region for population and investment, thereby enhancing the potential for urban-rural economic integration development.

In conclusion, these findings underscore the importance for local governments to carefully manage land fiscal policies to mitigate negative impacts on economic integration while fostering a conducive environment for urban-rural integration across various dimensions.

## Robustness checks

To ensure the credibility and reliability of the findings, given that baseline regression results may be influenced by factors such as sample selection bias, variable selection errors, and endogeneity issues, this study employed multiple robustness testing methods. First, to mitigate potential endogeneity issues, the system GMM estimation was applied, using instrumental variables to replace endogenous variables and reduce potential bias. Second, to control for biases that may arise from sample heterogeneity, this study excluded samples from municipalities and re-ran the regression analysis. Third, the study replaced the measurement method of the dependent variable to verify the consistency of the regression results under different variable definitions, thereby improving the rigor of the analysis. Lastly, to further test the robustness of the findings, this study introduced lagged explanatory variables to examine the impact of dynamic effects on the regression results.

1. **System GMM estimation.**

The GMM model is a commonly used estimation method to address endogeneity issues. Particularly in panel data analysis, urban-rural integration development may be influenced by other factors such as regional economic conditions, cultural traditions, and others, which might also have a bidirectional causal relationship with local governments' land transfer behaviors. Using conventional OLS or fixed effects models could lead to biased estimation results. The GMM estimation method, by employing instrument variables, effectively alleviates endogeneity issues, thus providing more reliable estimation results. Therefore, to obtain relatively unbiased and consistent estimation results, this study employs the system GMM estimation method to empirically examine the relationship between local governments' land transfer behaviors and urban-rural integration development (see Table 9). The regression results in Column (1) of Table 9 confirm that land price competition has a significantly positive effect on urban-rural integrated development at the 1% significance level, consistent with theoretical expectations. Column (2) further reveals a significant inverted-U-shaped relationship, indicating that while moderate land price competition promotes urban-rural integration, excessive competition may eventually weaken its positive effects. Additionally, Column (3) shows that fiscal reliance on land revenue has a significantly negative impact at the 10% level, reinforcing the robustness of our earlier findings. These results collectively validate our core conclusions regarding the differential effects of land market policies across regions.

2. **Excluding municipalities.**

Considering that municipalities directly under central government administration may exhibit significant differences in urban-rural integration development due to their economic characteristics, this study excludes sample data from Beijing,

**Table 9. System GMM estimation test results.**

|  | (1) | (2) | (3) |
|---|---|---|---|
|  | Degree | Degree | Degree |
| L.Degree | 0.9705*** | 0.9708*** | 0.9974*** |
|  | (0.0120) | (0.0120) | (0.0098) |
| lpc | 0.0004*** | 0.0007*** |  |
|  | (0.0002) | (0.0002) |  |
| (lpc)2 |  | −0.0007* |  |
|  |  | (0.0003) |  |
| lfd |  |  | −0.0023* |
|  |  |  | (0.0012) |
| jck | 0.0007** | 0.0007** | −0.0002 |
|  | (0.0003) | (0.0003) | (0.0003) |
| rlzb | 0.00005 | 0.00004 | 0.0001 |
|  | (0.00004) | (0.00004) | (0.0001) |
| zfgy | 0.0003 | 0.0003 | 0.0004 |
|  | (0.0003) | (0.0003) | (0.0005) |
| czh | 0.0043*** | 0.0045*** | 0.0028** |
|  | (0.0010) | (0.0010) | (0.0013) |
| dlmj | −0.0043*** | −0.0043*** | −0.0022* |
|  | (0.0011) | (0.0011) | (0.0013) |
| AR(1) | 0.024 | 0.024 | 0.025 |
| AR(2) | 0.203 | 0.202 | 0.197 |
| Hansen | 0.142 | 0.120 | 0.805 |
| N | 3,934 | 3,934 | 3,934 |

Tianjin, Shanghai, and Chongqing for regression estimation. The results, as shown in Regression (1) of Table 10. The empirical results demonstrate that land price competition has a significantly positive effect on urban-rural integrated development at the 1% significance level, while fiscal reliance on land revenue exerts a significantly negative impact at the 10% level. These findings align with theoretical expectations, confirming the robustness of the baseline regression estimates.

### 3. Replacement of the measurement of the explanatory variables.

To verify the accuracy of the previous regression results, this study further employed principal component analysis (PCA) to re-measure the level of urban-rural integration and conducted regression analysis based on this revised measurement, with the results shown in Table 10, regression column (2). By extracting the core characteristic variables of urban-rural integration through PCA, this method reduces the impact of potential multicollinearity and measurement errors caused by variable selection, thereby making the measurement results more objective and comprehensive. The regression results indicate that the coefficient of land price competition is significantly positive at the 1% level, while the coefficient of fiscal reliance on land revenue is significantly negative at the 1% level. These findings are consistent with previous conclusions, further validating that local governments' land transfer behaviors exert robust and divergent effects on urban-rural integrated development.

### 4. Number of periods of lagged explanatory variables.

Considering that land development and utilization may have a certain lag effect, and that the impact of local governments' land transfer behaviors on urban-rural integration may exhibit a time delay, this study introduced one-period lagged

**Table 10. Robustness test results.**

| | (1) Excluding municipalities | | (2) Replacement of the measurement | | (3) One period behind | |
|---|---|---|---|---|---|---|
| | Degree | Degree | Degree2 | Degree2 | Degree | Degree |
| lpc | 0.0007*** | | 0.0080*** | | | |
| | (0.0003) | | (0.0025) | | | |
| lfd | | −0.0012* | | −0.0192*** | | |
| | | (0.0006) | | (0.0062) | | |
| L.lpc | | | | | 0.0008*** | |
| | | | | | (0.0003) | |
| L.lfd | | | | | | −0.0017*** |
| | | | | | | (0.0006) |
| jck | −0.0014*** | −0.0014*** | −0.0058* | −0.0061* | −0.0019*** | −0.0019*** |
| | (0.0003) | (0.0003) | (0.0032) | (0.0032) | (0.0003) | (0.0003) |
| rlzb | −0.0001 | −0.0001 | −0.0007 | −0.0007 | −0.0001 | −0.0001 |
| | (0.0001) | (0.0001) | (0.0005) | (0.0005) | (0.0001) | (0.0001) |
| zfgy | −0.0021*** | −0.0022*** | −0.0397*** | −0.0401*** | −0.0017** | −0.0017** |
| | (0.0007) | (0.0007) | (0.0069) | (0.0069) | (0.0007) | (0.0007) |
| czh | −0.0036** | −0.0039** | 0.3653*** | 0.3617*** | −0.0099*** | −0.0103*** |
| | (0.0018) | (0.0018) | (0.0178) | (0.0177) | (0.0021) | (0.0021) |
| dlmj | −0.0060*** | −0.0059*** | −0.0593*** | −0.0577*** | −0.0073*** | −0.0071*** |
| | (0.0008) | (0.0008) | (0.0077) | (0.0077) | (0.0008) | (0.0008) |
| _cons | 0.0788*** | 0.0835*** | −0.3916*** | −0.3361*** | 0.1093*** | 0.1150*** |
| | (0.0074) | (0.0073) | (0.0715) | (0.0700) | (0.0084) | (0.0082) |
| N | 4155 | 4155 | 4215 | 4215 | 3934 | 3934 |
| r2 | 0.5664 | 0.5659 | 0.8002 | 0.8002 | 0.5985 | 0.5981 |
| Year fixed | Yes | Yes | Yes | Yes | Yes | Yes |
| City fixed | Yes | Yes | Yes | Yes | Yes | Yes |

explanatory variables to further validate the robustness of the empirical results. The regression results are shown in Table 10, regression column (3). Here, L.lpc and L.lfd respectively represent the one-period lagged values of land price competition and land fiscal dependence. The regression results reveal two key insights: The coefficient of land price competition is significantly negative at the 1% level, suggesting its role in promoting urban-rural integrated development exhibits a persistent influence over time rather than being limited to short-term effects; The coefficient of fiscal reliance on land revenue remains significantly negative at the 1% level, further confirming earlier findings that excessive dependence on land finance inhibits urban-rural integration—a result demonstrating robust consistency across analyses.

## Mechanism testing

In order to test the role of labor transfer effect, public service supply effect and industrial structure upgrading effect in the process of the influence of local government land grant behavior on urban-rural integration development, based on equations (2), (3) and (4), the moderating effect model is used to identify the interaction effects brought by labor transfer, public service supply and industrial structure upgrading, and the regression results are shown in Table 11. The results from regression (1) in Table 11 indicate that the interaction term between labor mobility and land price competition is significantly positive, and the coefficient of land price competition itself is also significantly positive. This suggests that labor mobility strengthens the positive effect of land price competition on urban-rural integrated development, thereby validating

**Table 11. Moderated effects test results.**

| | (1) | | (2) | | (3) | |
|---|---|---|---|---|---|---|
| | Degree | | Degree | | Degree | |
| lpc | 0.0009*** | | 0.0003 | | −0.0004 | |
| | (0.0003) | | (0.0002) | | (0.0003) | |
| labor | 4.68e-06 | −6.97e-06 | | | | |
| | (0.00001) | (0.00002) | | | | |
| lpc*labor | 0.000025** | | | | | |
| | (0.000011) | | | | | |
| lfd | | −0.0015** | | 0.0007 | | −0.0017*** |
| | | (0.0007) | | (0.0006) | | (0.0007) |
| lfd*labor | | −0.00004 | | | | |
| | | (0.00004) | | | | |
| service | | | 0.0258*** | 0.0282*** | | |
| | | | (0.0009) | (0.0007) | | |
| lpc*service | | | 0.0002 | | | |
| | | | (0.0003) | | | |
| lfd*service | | | | 0.0110*** | | |
| | | | | (0.0010) | | |
| cyjg | | | | | 0.0554*** | 0.0524*** |
| | | | | | (0.0046) | (0.0049) |
| lpc*cyjg | | | | | 0.0335*** | |
| | | | | | (0.0015) | |
| lfd*cyjg | | | | | | 0.0087** |
| | | | | | | (0.0044) |
| jck | −0.0013*** | −0.0013*** | −0.0006* | −0.0006** | −0.0010*** | −0.0013*** |
| | (0.0004) | (0.0003) | (0.0003) | (0.0003) | (0.0003) | (0.0003) |
| rlzb | −0.0001 | −0.0001 | −0.0001*** | −0.0001*** | −0.0001 | −0.0001** |
| | (0.0001) | (0.0001) | (0.0001) | (0.0000) | (0.0001) | (0.0001) |
| zfgy | −0.0030*** | −0.0031*** | −0.0031*** | −0.0032*** | −0.0027*** | −0.0029*** |
| | (0.0008) | (0.0008) | (0.0006) | (0.0006) | (0.0007) | (0.0007) |
| czh | −0.0052*** | −0.0056*** | 0.0097*** | 0.0091*** | −0.0021 | −0.0095*** |
| | (0.0019) | (0.0019) | (0.0017) | (0.0017) | (0.0019) | (0.0019) |
| dlmj | −0.0076*** | −0.0075*** | −0.0042*** | −0.0036*** | −0.0057*** | −0.0077*** |
| | (0.0008) | (0.0008) | (0.0007) | (0.0007) | (0.0008) | (0.0008) |
| _cons | 0.0926*** | 0.0928*** | 0.0425*** | 0.0439*** | 0.0794*** | 0.1106*** |
| | (0.0076) | (0.0076) | (0.0067) | (0.0066) | (0.0074) | (0.0077) |
| N | 4215 | 4215 | 4215 | 4215 | 4215 | 4215 |
| r2 | 0.5629 | 0.5617 | 0.6779 | 0.6869 | 0.6228 | 0.5743 |
| Year fixed | Yes | Yes | Yes | Yes | Yes | Yes |
| City fixed | Yes | Yes | Yes | Yes | Yes | Yes |

Hypothesis H2. The results from regression (2) in Table 11 show that the interaction term between public service provision and land price competition is not statistically significant, which contradicts Hypothesis H3. The results from regression (3) in Table 11 reveal that the interaction terms between industrial structure upgrading and both land price competition and land fiscal dependence are significantly positive, confirming Hypothesis H4.

## Discussion

The integration of urban and rural development is a crucial measure for advancing rural revitalization, with local government land transfer practices serving as a key link in promoting this integration. Based on panel data from 281 prefecture-level and above cities in China, this study constructs a two-way fixed effects model to categorize local government land transfer behaviors into two dimensions: low-price industrial land transfers and high-price commercial and residential land transfers, integrating them into a unified theoretical framework. The study systematically evaluates the impact of local government land transfer behaviors on urban-rural integration and conducts an empirical analysis of their effects and mechanisms. The research selects land price competition and fiscal dependence on land as core explanatory variables, employs the entropy method to calculate a comprehensive index of urban-rural integration as the dependent variable, and controls for variables such as the degree of openness, human capital level, and government intervention. In terms of mechanism testing, a moderating effects model is used to analyze the moderating effects of labor migration, public service provision, and industrial structure upgrading. The study also examines heterogeneous impacts across regions, administrative levels, and dimensions of urban-rural integration through subgroup regressions. Robustness checks are conducted using methods such as system GMM estimation, alternative measures of the dependent variable, and lagged explanatory variables. The key findings are as follows: (1) Among local government land transfer practices, land price competition exerts a significant positive effect on urban-rural integration while simultaneously exhibiting an inverted U-shaped relationship with it. (2) Fiscal dependence on land, however, inhibits urban-rural integration. (3) The impact of local government land transfer behavior on urban-rural integration exhibits heterogeneity across regions, administrative levels, and different dimensions of integration. (4) Mechanism analysis reveals that labor migration, public service provision, and industrial structure upgrading serve as critical channels through which local governments' land transfer behaviors influence urban-rural integration. Specifically, Labor migration and industrial structure upgrading strengthen the positive effect of land price competition; Public service provision and industrial structure upgrading mitigate the negative impact of fiscal dependence on land.

Compared to prior studies, this paper draws extensively on key insights from the literature regarding the relationship between land markets—such as land resource misallocation [10] and factor mobility [11]—and urban-rural integration. It also references empirical methods for measuring urban-rural integration levels, land marketization, and econometric model construction [51], while extending the theoretical framework on how land market reforms shape urban-rural dynamics [24]. Our findings confirm that land price competition positively contributes to urban-rural integration, whereas fiscal dependence on land exerts a negative effect. However, some scholars argue that land finance serves as a crucial engine for regional economic development [59]. Thus, dismissing its role entirely would be unbalanced; instead, efforts should focus on diversifying local government revenue sources to gradually reduce reliance on land finance and achieve sustainable urban-rural integration. Notably, this study exhibits significant distinctions from existing research in the following aspects: First, regarding research perspective, prior literature predominantly focuses on the impact of land markets per se or land resource allocation efficiency on urban-rural integration [29]. In contrast, this paper centers on local governments' land conveyance behaviors, internalizing them into two dimensions—land price competition and fiscal dependency on land revenue—thereby offering deeper insights into the mechanisms through which local governments' land transactions influence urban-rural integration. Second, existing studies posit that high-quality public service provision fosters a conducive environment for enterprises and investors, facilitating the entry and agglomeration of external firms and thereby promoting regional industrial upgrading and economic development [60]. However, our analysis reveals that public service provision does not exhibit a significant positive moderating effect in the relationship between local governments' land conveyance behaviors and urban-rural integration. A plausible explanation lies in the fact that while local governments' low-price industrial land transfers may attract more enterprises and capital, stimulating economic growth and employment opportunities, they simultaneously necessitate increased public service investments (e.g., infrastructure) to sustain such development. Yet, the public services provided may fail to adequately support the ancillary needs of enterprises attracted

by underpriced industrial land, thereby diminishing the anticipated reinforcing effect of public services on urban-rural integration under conditions of land price competition. Conversely, the interaction term between public service provision and fiscal dependency on land revenue shows a significantly positive coefficient. This suggests that enhancing public service levels—such as infrastructure and social security investments—can facilitate cross-regional flows of labor and other resources. Such improvements effectively mitigate adverse effects stemming from fiscal reliance on land revenue (e.g., soaring housing prices, industrial rigidity, and homogenization), thereby weakening its negative impact on urban-rural integration.

Furthermore, while this study systematically evaluates the impact of local governments' land conveyance behaviors on urban-rural integration, it may still have the following limitations: (1) In constructing the urban-rural integration development index system, due to the challenges of measurement and processing complexity, this paper, drawing on existing literature, only selected partial data from sources such as statistical yearbooks, prefecture-level city statistical yearbooks, the China Urban and Rural Construction Database, and the China Regional Economic Database. This approach may introduce slight deviations from the actual level of urban-rural integration. Additionally, the selected indicators are limited to quantitative metrics from statistical yearbooks and similar sources, excluding qualitative factors such as government institutions, governance structures, and officials' service awareness. (2) Although variables such as the degree of openness to the outside world and human capital levels were controlled for, other potential influencing factors—such as local policy preferences, historical development trajectories, and natural geographical conditions—may have been overlooked in their impact on urban-rural integration. (3) This study only examined the moderating effects in the relationship between local governments' land conveyance behaviors and urban-rural integration, without analyzing whether these behaviors indirectly influence urban-rural integration through mediating variables.

## Conclusions

This study employs panel data from 281 prefecture-level and above cities in China, utilizing bidirectional fixed-effects models and moderated mediation models to analyze local governments' land conveyance practices. These practices are categorized into two dimensions—low-priced industrial land transfers and high-priced commercial/residential land transfers—and examined within a unified theoretical framework. The research systematically establishes the causal relationship between local governments' land conveyance strategies and urban-rural integrated development, while empirically analyzing their effects and underlying mechanisms. However, several limitations remain: (1) potential minor discrepancies between the constructed urban-rural integration index system and actual integration levels; (2) possible omission of other control variables that may influence urban-rural development; and (3) insufficient examination of whether moderating variables indirectly affect urban-rural integration through mediating variables.

Based on the research findings of this study, the following policy recommendations are proposed:

Optimizing Industrial Land Pricing and Reducing Dependency on Land Finance: Considering the "inverted U-shaped" relationship between low-priced industrial land transfers by local governments and urban-rural integration, excessively low land transfer prices can lead to regional competition issues that hinder integration. Conversely, excessively high prices can diminish the positive effects of land price competition. Therefore, local governments need to find a reasonable balance in the pricing of industrial land transfers. To reduce land costs for enterprises and attract high-quality external enterprises, local governments can explore diversified land supply mechanisms, such as land leasing, "rent-first, sell-later" schemes, and long-term leases. These approaches not only alleviate the initial investment pressure on enterprises but also allow for more flexible adjustments to the efficiency of land resource allocation. Based on this, governments can guide high-quality enterprises to form industrial chain effects through agglomeration, promote the scaled development of urban economies, and leverage the city's role in driving rural areas. This facilitates the optimized flow of economic factors between urban and rural areas, achieving the long-term sustainability of urban-rural integration.At the same time, local governments should gradually reduce their heavy reliance on land-based fiscal revenues and promote the diversification

of fiscal revenue sources. For example, governments can develop emerging industries, expand regional financing channels, and advance public-private partnership (PPP) models to increase fiscal revenue sources. Additionally, strict oversight of land transfer revenue utilization is necessary, with funds prioritized for urban-rural infrastructure development, the equalization of public services, and rural revitalization initiatives. This helps avoid the negative effects of land-based finance, such as high housing prices and homogeneous industrial structures, which hinder urban-rural integration. In this way, a solid foundation for coordinated urban-rural economic development can be established.

Addressing Regional, Administrative, and Dimensional Heterogeneities: In regions like the eastern and central areas, and in cities at higher administrative levels, enhancing infrastructure support and policy incentives for external enterprises in low-priced industrial land areas is crucial. This effort lowers entry barriers and resource input costs for enterprises, maximizing economies of scale to promote urban support for rural areas and break down urban-rural barriers. Simultaneously, efforts should align with the threshold and innovation effects brought by land price competition to drive economic integration. Regarding land fiscal dependence, western regions and cities with lower administrative levels need to reduce biased expenditures driven by land-based fiscal revenues and reallocate these resources toward infrastructure, healthcare, education, and other public services. Increasing the supply of affordable housing is also essential to alleviate pressure on housing prices. For regions highly dependent on land-based fiscal revenues, strategies such as developing diversified fiscal revenue sources, promoting characteristic and green industries, optimizing land revenue utilization models, and establishing interregional fiscal balance mechanisms can gradually reduce over-reliance on land transfer revenues and foster urban-rural integration.

Leveraging Labor Migration and Industrial Structure Upgrading: Highlighting the enhanced role of labor migration and industrial structure upgrading in promoting urban-rural integration through local government's low-priced industrial land transfers. Additionally, focusing on public service provision and industrial structure upgrading in high-priced commercial and residential land transfers to mitigate their inhibitory effects on urban-rural integration. To achieve these goals, on one hand, local governments should increase investment in regional infrastructure development, improving transportation networks, information and communication systems, and other physical infrastructure. At the same time, optimizing public services such as social security and housing benefits can enhance cities' inclusiveness and attractiveness to rural labor forces, facilitating the free flow of labor factors between urban and rural areas and promoting optimized resource allocation. On the other hand, local governments should focus on improving human capital levels by strengthening vocational education and skills training. This enhances the overall quality and adaptability of the labor force, while policy guidance can direct labor flows toward advantageous and emerging industries, achieving regional industrial structure optimization and upgrading. These measures strengthen economic endogenous growth and further promote urban-rural integration.

## Supporting information

**S1 Data.**
(XLSX)

## Acknowledgments

The authors extend their heartfelt appreciation to the esteemed peer reviewers for their invaluable reviews and constructive comments, which significantly contributed to the refinement and enhancement of this study.

## Author contributions

**Data curation:** Jin Xie.

**Formal analysis:** Bin Peng.

**Funding acquisition:** Long Zeng.

**Methodology:** Bin Peng.

**Software:** Bin Peng.

**Supervision:** Long Zeng.

**Visualization:** Jin Xie.

**Writing – original draft:** Bin Peng.

**Writing – review & editing:** Long Zeng.

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
