## [Decision Letter · Decision Letter 0]

Dear Dr. Peng,

Thank you for submitting your manuscript to PLOS ONE. After careful consideration, we feel that it has merit but does not fully meet PLOS ONE’s publication criteria as it currently stands. Therefore, we invite you to submit a revised version of the manuscript that addresses the points raised during the review process.

-------------------

**In addition to reviewers' comments, please consider the following comments from academic editor while revising your manuscript:**

Novelty and Clarity of Research Gap: 

The study identifies a critical research gap by focusing on the impact of local government land ‎transfer practices on urban-rural integration development, emphasizing mechanisms such as ‎land price competition and fiscal dependence. While this topic is significant and addresses a ‎policy-relevant issue in the context of China, the novelty is somewhat limited due to the ‎extensive prior work on land use and urbanization. The study's differentiation could be ‎improved by further elaborating on how its approach surpasses existing research.‎

Literature Review: 

The paper provides a robust review of relevant literature, delineating prior studies on urban-‎rural integration and the role of land markets. However, it could better integrate more diverse ‎international perspectives to highlight potential parallels or deviations in global contexts. ‎Additionally, while the hypotheses are well-articulated, their connection to the literature could ‎be more explicitly justified.‎

Methodology: 

The use of panel data from 281 cities and bidirectional fixed effects modeling is appropriate ‎and rigorous. However, the paper could provide more clarity on the rationale for selecting ‎specific control variables and the measurement of constructs like "urban-rural integration ‎development." The robustness checks strengthen the validity, yet the discussion of potential ‎limitations in variable definitions or endogeneity concerns requires expansion.‎

Statistical Analysis: 

The statistical results are presented systematically, with clear interpretation of coefficients and ‎significant effects. However, the justification for the inverted U-shaped relationship between ‎land price competition and integration development could be more thoroughly grounded in ‎empirical patterns or theoretical reasoning.‎

Theoretical Contribution: 

The study makes a meaningful theoretical contribution by incorporating labor transfer, public ‎services, and industrial structure upgrading into a unified framework. Nonetheless, it would ‎benefit from a deeper discussion of how these mechanisms extend or challenge existing ‎theories on urban-rural dynamics.‎

Practical Applications: 

While the paper provides actionable insights, such as the need for regional customization of ‎land policies, its recommendations could be more nuanced. For instance, proposing strategies ‎for regions with high land finance dependence beyond reducing reliance could add depth to the ‎practical implications.‎

We look forward to receiving your revised manuscript.

Kind regards,

Amar Razzaq, PhD

Academic Editor

PLOS ONE

Journal Requirements:

“This research was made possible through the generous support of the National Natural Science Foundation of China (72303062; 72304095), the Humanities and Social Sciences Foundation of the Ministry of Education (22YJC790007), and the Natural Science Foundation of Hunan Province (2021JJ40263). “

Reviewers' comments:

Reviewer's Responses to Questions

**Comments to the Author**

1. Is the manuscript technically sound, and do the data support the conclusions?

Reviewer #1: Yes

Reviewer #2: Yes

2. Has the statistical analysis been performed appropriately and rigorously?

Reviewer #1: Yes

Reviewer #2: No

3. Have the authors made all data underlying the findings in their manuscript fully available?

Reviewer #1: Yes

Reviewer #2: No

4. Is the manuscript presented in an intelligible fashion and written in standard English?

Reviewer #1: Yes

Reviewer #2: No

Reviewer #1: The study is good effort and well organized and well written. However, the anther needs to explain practical and theoretical implications of the empirical findings study. Also explain the significance the study.

Reviewer #2: The review process of the manuscript is completed.

In my opinion, the justification and theoretical foundations of the research have been well explained. However, this study is full of shortcomings in econometric models.

You have justified some of the findings. Provide references for these sentences. For example, in part 4.2.1 Regression results and analysis by region, some sentences about the possible reasons for the difference in results in different regions need to be referenced.

In Table 6, the column of social criteria is repeated twice. Please correct.

In all the tables, tell what the numbers in parentheses represent.

The coefficients obtained in the estimated models are numerically very low. This issue is probably related to the different units of the variables. Check this item and check by changing the unit of variables.

On the other hand, somewhere in the text, it is stated that it is taken from the variables ln. While some variables are in the form of percentages or ratios and cannot be taken as ln.

In panel data, the unit root test and the presence or absence of cointegration in the long run between variables are necessary.

Why is the gmm model used? This model is a dynamic method and there is no evidence of this in the results table. And of course, why are there only three degrees in gmm results in table 7?

List the units of the variables in a column in Table 2.

In line 256, the notation error term is reported incorrectly.

**Do you want your identity to be public for this peer review?** For information about this choice, including consent withdrawal, please see our Privacy Policy

Reviewer #1: No

Reviewer #2: No

---

## [Author Response · Author response to Decision Letter 1]

12 Jan 2025

Dear Editor and Reviewers,

Thanks very much for taking your time to review this manuscript. we really appreciate all your comments and suggestions! We provide a detailed response to each comment raised by the reviewers and outline the revisions made accordingly. In addition, the modifications we have made have been highlighted in red font in the re-submitted files.The relevant files have been uploaded to the system.

---

## [Decision Letter · Decision Letter 1]

Dear Dr. Peng,

Thank you for submitting your manuscript to PLOS ONE. After careful consideration, we feel that it has merit but does not fully meet PLOS ONE’s publication criteria as it currently stands. Therefore, we invite you to submit a revised version of the manuscript that addresses the points raised during the review process.

We look forward to receiving your revised manuscript.

Kind regards,

Amar Razzaq, PhD

Academic Editor

PLOS ONE

Reviewers' comments:

Reviewer's Responses to Questions

**Comments to the Author**

Reviewer #2: All comments have been addressed

Reviewer #3: All comments have been addressed

Reviewer #4: (No Response)

Reviewer #5: (No Response)

2. Is the manuscript technically sound, and do the data support the conclusions?

Reviewer #2: Yes

Reviewer #3: Yes

Reviewer #4: Yes

Reviewer #5: No

3. Has the statistical analysis been performed appropriately and rigorously?

Reviewer #2: Yes

Reviewer #3: Yes

Reviewer #4: Yes

Reviewer #5: Yes

4. Have the authors made all data underlying the findings in their manuscript fully available?

Reviewer #2: Yes

Reviewer #3: Yes

Reviewer #4: Yes

Reviewer #5: Yes

5. Is the manuscript presented in an intelligible fashion and written in standard English?

Reviewer #2: Yes

Reviewer #3: Yes

Reviewer #4: Yes

Reviewer #5: No

Reviewer #2: The revisions have been well done by the author. The address of these changes is well highlighted in the text.

Reviewer #3: The authors have made substantial efforts to incorporate the feedback provided in the first round of reviews. The revised version of the manuscript demonstrates significant improvements in methodological clarity, theoretical grounding, and empirical robustness. The responses provided to the reviewers' comments indicate a serious and constructive engagement with the feedback, leading to a more refined and academically rigorous paper.

Strengths of the Revised Manuscript:

- Thorough incorporation of reviewer feedback – The authors have made significant efforts to address the reviewers’ comments, particularly by clarifying their research framework, refining their data presentation, and strengthening their discussion of mechanisms.

- Clearer methodology and robustness checks – The manuscript now includes a more detailed description of the econometric models used, as well as additional robustness tests, which enhance the reliability of the results.

- Improved discussion of policy implications – The authors have strengthened their discussion on the practical implications of their findings, offering insights for policymakers and urban planners.

- Better presentation of empirical results – The tables and figures have been revised, improving readability and making the results easier to interpret.

Reviewer #4: The article deals with an interesting and current topic. It discusses the relationship between land prices and regional development. The topic is suitable for publication in a journal. However, I have a few comments on the processing. In my opinion, the literature review is not sufficiently processed, even though it is a case study, it would be appropriate to supplement the review with the results of research on the given issue from other countries. I am not entirely sure whether the economic performance hypothesis and the Land finance hypothesis are appropriately formulated as hypotheses. After reading it, they are more like research questions. Because hypotheses should be based directly on the problem being solved and not refer to the authors. For H1: "Selling industrial land at low prices.... it is no longer stated what the low prices are and on what basis they were defined as low prices. And also "high industrial land prices" - on what basis are high prices defined? Why are relatively old data used? (The study's dataset spans from 2003 to 2017). There are no newer ones available, since the price of land is a dynamic variable and what was current in 2017 may not be current in 2025. It would be appropriate to add a period of at least 5 years. Or to justify well why this particular time series of data is used. Is it possible to make recommendations for the current period based on relatively old data? On what basis was the sample of cities selected? Is it a statistically significant sample? In the conclusion, the research limits are missing.

Reviewer #5: Here are the strengths of the manuscript: the issue is interesting and eligible for academic discussion, with major variables identified in the numeric analysis. However, there are several areas that need improvement:

1. The manuscript lacks clarity in its preparation.

2. It does not provide sufficient depth and breadth in the theoretical and conceptual framework.

3. The literature review section needs to be reworked to address topical issues and include relevant empirical studies.

4. The research methodology (approach, design, data gathering tools, sampling, analysis, etc.) is not adequately described.

5. The research discussion is not supported by empirical evidence or other literature; it lacks triangulation techniques.

6. The study lacks qualitative data sources and their analysis, focusing solely on numerical data. Therefore, the manuscript is not suitable for policy research.

7. The manuscript contains redundancy of ideas throughout the paper.

8. The title of the manuscript requires modification.

9. It lacks coherent technical procedures

**Do you want your identity to be public for this peer review?** For information about this choice, including consent withdrawal, please see our Privacy Policy

Reviewer #2: No

Reviewer #3: **Yes: ** Diogo Henrique Helal

Reviewer #4: No

Reviewer #5: No

---

## [Author Response · Author response to Decision Letter 2]

27 Apr 2025

Manuscript ID:PONE-D-24-33248

Title: Research on the Impact and Mechanism of Local Government Land Transfer Behavior on Urban Rural Integration Development——Perspectives based on land price competition and land finance dependence

Authors name:Long Zeng , Bin Peng' * and Jin Xie

Response to Reviewers

Dear Editor and Reviewers,

Thanks very much for taking your time to review this manuscript. we really appreciate all your comments and suggestions! Below, we provide a detailed response to each comment raised by the reviewers and outline the revisions made accordingly. In addition, the modifications we have made have been highlighted in red font in the re-submitted files.

Reply to the comments of Reviewer 1

1.Please expand on the practical and theoretical implications, and clarify the study’s significance

A Thank you very much for your suggestions. In the introduction section, we have elaborated on the theoretical and practical significance of this study in the current research field, rather than providing a general overview. The excerpt is as follows:

“The potential marginal contributions of this study are as follows: First, this paper is the first to systematically elucidate the impact mechanism of local governments' land transfer behavior on urban-rural integration, filling a gap in the existing literature. By constructing a unified theoretical framework, this study deeply explores the internal logic of how land transfer behavior influences urban-rural integration through channels such as labor migration, public service provision, and industrial structure upgrading. This provides a novel perspective for the theoretical discourse on the relationship between land transfer behavior and urban-rural integration.Second, this research provides theoretical support and practical guidance for local governments to optimize policies in the process of land transfer. Particularly in the context of promoting urban-rural integration, the findings of this study help local governments better understand the multidimensional impacts of land transfer behavior and thereby formulate more scientific land transfer and fiscal policies. Furthermore, the policy recommendations proposed in this study can assist governments in rationally adjusting land transfer practices to facilitate labor migration, enhance public service provision, and promote industrial structure upgrading. In doing so, this study aims to foster urban-rural integration, reduce the urban-rural gap, and achieve coordinated economic and social development.”Please refer to lines 109 to 121 in the article for the specific details.

Reply to the comments of Reviewer 2

1.You have justified some of the findings. Provide references for these sentences. For example, in part 4.2.1 Regression results and analysis by region, some sentences about the possible reasons for the difference in results in different regions need to be referenced.

A Thank you very much for your suggestions. Regarding the possible reasons for the differences in results across regions, we have further cited relevant literature in the paper to supplement the analysis of the causes of regional disparities and emphasized how these factors affect urban-rural integration, thereby enhancing the theoretical depth and practical significance of the study. The excerpt is as follows:

“In China’s eastern and central regions, industrial development occurred relatively early, and economic development has gradually transitioned from traditional heavy industries to high-tech and service industries[53]. The relatively high labor costs in these regions have made low-priced industrial land transfers less effective in attracting enterprises, thereby diminishing their impact on urban-rural integration.”Please refer to lines 396 to 399 in the article for the specific details.

“From the regression results on land fiscal dependence, it is evident that the dependence on land finance has a significant negative effect on urban-rural integration in western China, while it exhibits a significant positive effect in the central region and no significant impact in the eastern region. The potential reasons for these findings include the following: the western region’s underdeveloped economy, combined with low household income and consumption levels, reduces enterprises' willingness to engage in diversified investments[54]. This situation exacerbates issues such as industrial structure rigidity and homogeneity caused by local governments' high-priced transfers of commercial and residential land, ultimately hindering urban-rural integration.”Please refer to lines 400 to 406 in the article for the specific details.

2.In Table 6, the column of social criteria is repeated twice. Please correct.

A Thank you very much for your thorough review. First, due to the addition of relevant tables presenting unit root and cointegration test results during the revision process, Table 6 has now been renumbered as Table 8. Second, regarding the issue of the social integration dimension being listed twice, this is due to the limitations of table length. In this paper, Table 8 presents the social integration dimension in two separate sections: the upper section displays the regression results of land price competition on urban-rural social integration, while the lower section shows the regression results of land fiscal dependence on urban-rural social integration. To avoid any misunderstanding, we will include a clear explanation of this layout in the table notes to ensure that readers can accurately interpret the content and logical structure of the table. The specific excerpt is as follows:

“Due to table length limitations, Table 8 divides the social integration dimension into two parts: the upper section presents the regression results of land price competition on urban-rural social integration, while the lower section displays the regression results of land fiscal dependence on urban-rural social integration.”Please refer to lines 466 to 468 in the article for the specific details.

3.In all the tables, tell what the numbers in parentheses represent.

A Thank you very much for your suggestion. In the notes under Table 3, we have provided a detailed explanation of the meaning of the numbers in parentheses in the table. The excerpt is as follows:

“***, **, and * indicate significance levels at 1%, 5%, and 10%, respectively. The numbers in parentheses represent standard errors. The same applies to the following tables.”Please refer to lines 379 to 380 in the article for the specific details.

4.The coefficients obtained in the estimated models are numerically very low. This issue is probably related to the different units of the variables. Check this item and check by changing the unit of variables.

A Thank you very much for your thorough review. As the reviewer mentioned, this paper identified the issue of relatively low estimated coefficient values during the initial drafting stage. We also attempted various solutions, such as changing the units of variables, to address this issue. For example, the paper conducted a re-analysis of the baseline regression after modifying the units of the variables for land price competition and land fiscal dependence. The results are as follows: the estimated coefficients remained relatively low in numerical value.We believe that, although adjusting variable units can alter the numerical value of the coefficients, it does not change the actual impact of the independent variables on the dependent variables. Therefore, we consider these results to be credible and have retained the original regression results without further adjustment.

(1) (2) (3) (4)

Degree Degree Degree Degree

lpc 0.0009*** 0.0009*** 0.0009***

(0.0003) (0.0003) (0.0003)

lfd -1.42e-07 ** -1.56e-07**

(6.73e-08 ) (6.73e-08)

lpc2� -0.0002

(0.0002)

jck -0.0013*** -0.0013*** -0.0013*** -0.0013***

(0.0003) (0.0003) (0.0003) (0.0003)

rlzb -0.0001 -0.0001 -0.0001 -0.0001

(0.0001) (0.0001) (0.0001) (0.0001)

zfgy -0.0030*** -0.0031*** -0.0031*** -0.0030***

(0.0008) (0.0008) (0.0008) (0.0008)

czh -0.0052*** -0.0056*** -0.0052*** -0.0053***

(0.0019) (0.0019) (0.0019) (0.0019)

dlmj -0.0077*** -0.0075*** -0.0076*** -0.0077***

(0.0008) (0.0008) (0.0008) (0.0008)

_cons 0.0961*** 0.0938*** 0.0969*** 0.0968***

(0.0077) (0.0076) (0.0077) (0.0077)

N 4215 4215 4215 4215

r2 0.5624 0.5616 0.5630 0.5625

Year fixed Yes Yes Yes Yes

City fixed Yes Yes Yes Yes

5.On the other hand, somewhere in the text, it is stated that it is taken from the variables ln. While some variables are in the form of percentages or ratios and cannot be taken as ln.

A Thank you very much for your thorough review. This paper has provided further explanation regarding the logarithmic transformation of certain variables. The specific excerpt is as follows:

"In addition, to minimize heteroscedasticity and reduce the influence of outliers on the model, this study introduces certain variables into the equations in logarithmic form based on their magnitudes. For instance, variables such as land price competition, urbanization level, and infrastructure level are log-transformed to improve the model's fit and robustness."Please refer to lines 330 to 334 in the article for the specific details.

6.In panel data, the unit root test and the presence or absence of cointegration in the long run between variables are necessary.

A Thank you very much for your suggestion. In the empirical section of the paper, we have incorporated unit root and cointegration tests. The excerpt is as follows:

Panel Unit Root and Cointegration Tests

To avoid spurious regression, it is first necessary to perform unit root tests on the panel data to determine its stationarity. This study employs both the IPS test and LLC test, which are commonly used for homogeneous unit root testing, to examine the stationarity of each variable in the panel data. If both tests reject the null hypothesis of a unit root, it indicates that the variable is stationary. Conversely, if the null hypothesis of a unit root is accepted, it implies that the variable is non-stationary.

The test results, as shown in Table 3, indicate that for all variables, the first-order differenced results reject the null hypothesis of "the existence of a unit root" at the 1% significance level. This confirms that all variables are stationary, thereby ruling out the possibility of spurious regression. However, due to the potential instability of panel data, directly applying the ordinary least squares (OLS) method may still result in spurious regression. Therefore, a panel cointegration test is performed next to analyze whether the relevant variables exhibit cointegration relationships.

Table 3 Results of Unit Root Test for Variables

Variables IPS LLC Conclusion

D(Degree) Statistic -25.0291*** -16.7626*** stable

P 0.0000 0.0000

D(lpc) Statistic -34.3859*** -32.7703*** stable

P 0.0000 0.0000

D(lfd) Statistic -34.1674*** -36.0627*** stable

P 0.0000 0.0000

D(jck) Statistic -21.7426*** -5.0324*** stable

P 0.0000 0.0000

D(rlzb) Statistic -2.7442*** -63.9612*** stable

P 0.0030 0.0000

D(zfgy) Statistic -31.7085*** -28.4874*** stable

P 0.0000 0.0000

D(czh) Statistic -23.0573*** -1.5e+02*** stable

P 0.0000 0.0000

D(dlmj) Statistic -25.3438*** -31.6995*** 平稳

P 0.0000 0.0000

Note: D denotes first-order differencing, and *, **, *** represent significance levels of 10%, 5%, and 1%, respectively, at which the null hypothesis is rejected.

Based on the unit root test results, this study further conducts a cointegration test to verify the equilibrium relationship between variables. Specifically, the Pedroni test, Westerlund test, and Kao test are employed. The results indicate that the p-values of the test statistics are all less than 0.01. Therefore, the null hypothesis of no cointegration is rejected at the 1% significance level, suggesting the existence of a long-term stable cointegration relationship among the variables. This provides a solid foundation for conducting regression analysis.

Table 4 Cointegration test results

Inspection type Inspection Statistics Statistics P

Pedroni Modified Phillips-Perron t 30.0642 0.0000

Phillips-Perron t -21.0833 0.0000

Augmented Dickey-Fuller t -16.4603 0.0000

Westerlund Variance ratio 22.7037 0.0000

Kao Modified Dickey-Fuller t 8.5473 0.0000

Dickey-Fuller t 8.0558 0.0000

Augmented Dickey-Fuller t 13.9056 0.0000

Unadjusted modified Dickey-Fuller t 2.6583 0.0039

Unadjusted Dickey-Fuller t 1.6721 0.0472

Please refer to lines 338 to 358 in the article for the specific details.

7.Why is the gmm model used? This model is a dynamic method and there is no evidence of this in the results table. And of course, why are there only three degrees in gmm results in table 7?

A:Thank you very much for your suggestion. The rationale for using the GMM model lies in the fact that, although GMM is a dynamic method, it is also commonly used to address endogeneity issues. In panel data analysis, urban-rural integration development may be influenced by other factors such as regional economic conditions and cultural traditions, which might have a bidirectional causal relationship with local governments' land transfer behaviors. Using conventional OLS or fixed effects models may result in biased estimation results. By leveraging instrument variables, the dynamic GMM method effectively mitigates endogeneity problems, providing more reliable estimation results. Therefore, this study adopts the GMM model, drawing on established practices in the literature, to conduct robustness tests.

Additionally, the GMM test results are presented in only three columns for the following reason: based on the theoretical framework established earlier, this study employs the system GMM estimation to empirically test three aspects: (1) the impact of land price competition on urban-rural integration, (2) the impact of land fiscal dependence on urban-rural integration, and (3) the "inverted U-shaped" relationship observed between land price competition and urban-rural integration. Consequently, the GMM results are displayed in only three columns.To further clarify the rationale for using the GMM model, we will explicitly explain its usage in the paper and elaborate on all three columns of results in the discussion section. The excerpt is as follows:

"The GMM model is a commonly used estimation method to address endogeneity issues. Particularly in panel data analysis, urban-rural integration development may be influenced by other factors such as regional economic conditions, cultural traditions, and others, which might also have a bidirectional causal relationship with local governments' land transfer behaviors. Using conventional OLS or fixed effects models could lead to biased estimation results. The GMM estimation method, by employing instrument variables, effectively alleviates endogeneity issues, thus providing more reliable estimation results. Therefore, to obtain relatively unbiased and consistent estimation results, this study employs the system GMM estimation method to empirically examine the relationship between local governments' land transfer behaviors and urban-rural integration development (see Table 9).From column (1) in Table 9, it can be seen that the regression results for the core explanatory variable align with theoretical expectations, indicating that land price competition has a significant positive impact on urban-rural integration at the 1% significance level. Column (2) of Table 9 shows that land fiscal dependence has a significant negative impact on urban-rural integration at the 10% significance level. Column (3) of Table 9 further demonstrates that land price competition exhibits a significant "inverted U-shaped" relationship with urban-rural integration, which suggests that the regression results discussed earlier are robust."Please refer to lines 479 to 493 in the article for the specific details.

8.List the units of the variables in a column in Table 2.

A:Thank you for your suggestion. The units of variables were not listed in Table 2 primarily due to the calculation methods of certain indicators. On the one hand, some variables are derived as ratios or propo

---

## [Decision Letter · Decision Letter 2]

Dear Dr. Peng,

Thank you for submitting your manuscript to PLOS ONE. After careful consideration, we feel that it has merit but does not fully meet PLOS ONE’s publication criteria as it currently stands. Therefore, we invite you to submit a revised version of the manuscript that addresses the points raised during the review process.

We look forward to receiving your revised manuscript.

Kind regards,

Amar Razzaq, PhD

Academic Editor

PLOS ONE

Reviewers' comments:

Reviewer's Responses to Questions

**Comments to the Author**

Reviewer #4: All comments have been addressed

Reviewer #5: All comments have been addressed

2. Is the manuscript technically sound, and do the data support the conclusions?

Reviewer #4: Yes

Reviewer #5: Yes

3. Has the statistical analysis been performed appropriately and rigorously?

Reviewer #4: Yes

Reviewer #5: Yes

4. Have the authors made all data underlying the findings in their manuscript fully available?

Reviewer #4: Yes

Reviewer #5: Yes

5. Is the manuscript presented in an intelligible fashion and written in standard English?

Reviewer #4: Yes

Reviewer #5: Yes

Reviewer #4: The article deals with an interesting topic. However, according to the template, a literature review should be provided separately. The article addresses the issue in medias res without a theoretical background. It would be appropriate to supplement the theoretical background of the problem being addressed, how the given topic is being addressed abroad.

The article is appropriately processed from a methodological point of view, but the limits of the research are missing in the conclusion.

Reviewer #5: The author edited and modified the manuscript to enhance the logical soundness of the work, incorporating all the comments I provided. The aim of the study and the data support the conclusion; the statistical analysis was conducted rigorously, and all data are fully available in the manuscript, written in clear language. Due to my repeated reviews and professional evaluation, the manuscript addresses the comments forwarded and achieves the criteria for publication in PLOS ONE. Therefore, I have accepted the manuscript for publication.

**Do you want your identity to be public for this peer review?** For information about this choice, including consent withdrawal, please see our Privacy Policy

Reviewer #4: No

Reviewer #5: **Yes: ** Elias Munye Dagnew, Senior Lecturer of Regional and Local Development Studies at DebreMarkos University of Ethiopia, E-mail eliasmunye@gmail.com, elias_munye@dmu.edu.et, Phone:+251910559015

---

## [Author Response · Author response to Decision Letter 3]

27 May 2025

Reply to the comments of Reviewer 4

1.The article deals with an interesting topic. However, according to the template, a literature review should be provided separately.

A Thank you very much for your careful review. Based on your suggestion, this article has separated the literature review content from the current article, and the specific abstract is as follows:

“Literature Review

Under the urban-rural dual structure, cities typically offer more employment opportunities and higher wages, which incentivize the migration of labor and other production factors from rural to urban areas. However, with the evolving development environment between urban and rural regions, a reverse trend has emerged, where production factors, including labor, are increasingly flowing back from cities to rural areas. Urban-rural integrated development is widely regarded as a process that facilitates the rational flow and optimal allocation of production factors between urban and rural areas, thereby promoting economic integration and interactive growth. Existing literature has explored the conceptual framework and practical challenges of urban-rural integration from a theoretical perspective [5], while also identifying its driving mechanisms and pathways [6]. Empirical studies have further measured the level of urban-rural integration and examined its influencing factors from various dimensions [7], including economic growth [8], economic agglomeration [9], misallocation of urban-rural production factors [10], factor mobility [11], water quality improvement measures [12], new industrial cooperation models [13], and infrastructure development [14]. These studies often adopt analytical units at the provincial or prefecture-city level [15].

The academic community has deeply explored the relationship between land sales and urban-rural integration development. As a resource element and spatial carrier of urban and rural economic development, land has become a crucial force in the evolution of China's land system reform through market mechanisms [16]. Against the backdrop of a dual urban-rural land resource allocation system [17], urban-biased land acquisition methods [18], and government competition [19], land element allocation strategies and reforms have become important tools for local governments to develop regional economies [20-21].Overall, research on the relationship between land sales and urban-rural integration development can be divided into two main categories: Indirect Impact Studies: These studies examine how local government land sale practices indirectly influence urban-rural integration development by affecting related areas such as regional industrial structure upgrading [22], economic growth [23], urbanization development [24], urban-rural income gaps [25], Agricultural efficiency [26], sustainable development [27], and public services [28]. Direct Impact and Reform Path Studies: These studies explore the mechanisms [29-30], historical evolution, mechanism design [31], and institutional innovations [32] through which the Chinese land market impacts urban-rural integration development. On this basis, scholars further analyze the effects of land resource misallocation [11], land resource allocation efficiency [33], land use rates, and land transformation [34] on urban-rural integration development.

In summary, existing research has provided a significant theoretical and empirical foundation for this study. Existing literature predominantly focuses on the impacts of land markets, land resource allocation efficiency, and misallocation on urban-rural integration, while paying relatively little attention to the effects of local governments' land transfer behaviors on urban-rural integration. Furthermore, previous studies have not systematically conceptualized local governments' land transfer behaviors into the two dimensions of land price competition and land fiscal dependence to explore their impacts and heterogeneity concerning urban-rural integration. Additionally, the mechanisms through which local governments' land transfer behaviors influence urban-rural integration remain largely unexplored. The potential marginal contributions of this study are as follows: First, this paper is the first to systematically elucidate the impact mechanisms of local governments' land transfer behaviors on urban-rural integration, filling a gap in the existing literature. By constructing a unified theoretical framework, this study deeply examines how land transfer behaviors influence urban-rural integration through mechanisms such as labor migration, public service provision, and industrial structure upgrading. This provides a novel perspective for understanding the relationship between land transfer behaviors and urban-rural integration. Second, this study offers theoretical support and practical guidance for optimizing land transfer policies by local governments. In particular, in the context of promoting urban-rural integration, the findings of this paper help local governments better understand the multidimensional impacts of land transfer behaviors, enabling them to formulate more scientific land transfer and fiscal policies. Furthermore, the policy recommendations provided in this study can help governments rationally adjust their land transfer practices to promote labor migration, enhance public service provision, and facilitate industrial structure upgrading, thereby fostering urban-rural integration, reducing the urban-rural gap, and achieving coordinated economic and social development.”

For specific details, please refer to lines 93 to 141 of the manuscript.

2.The article addresses the issue in medias res without a theoretical background. It would be appropriate to supplement the theoretical background of the problem being addressed, how the given topic is being addressed abroad.

A Thank you very much for your suggestion. This article has made modifications to the introduction section and supplemented the theoretical background of the research content and empirical evidence on how foreign countries handle it. The specific summary is as follows

“The integrated development of urban and rural areas serves as a fundamental pathway for China's comprehensive rural revitalization and a crucial measure to dismantle the urban-rural duality and achieve common prosperity. As explicitly stated in the report of the 20th National Congress of the Communist Party of China, efforts should be made to promote urban-rural integration and coordinated regional development, facilitate the circulation of the national economy, and establish a new development dynamic between urban and rural areas. Consequently, how to achieve integrated urban-rural development has become a significant proposition in China's current phase of new development. From a theoretical perspective, research on urban-rural integrated development is rooted in theories of urban-rural relations, spatial economics, and institutional economics. Lewis's dual-sector model first revealed the economic structure of urban-rural segmentation in developing countries, highlighting the coexistence of traditional agricultural sectors and modern industrial sectors. The Todaro model further analyzed the economic motivations behind rural-urban migration, emphasizing the role of expected income differentials in labor mobility. Meanwhile, new economic geography explains, from a spatial perspective, how industrial agglomeration and dispersion shape urban-rural relations. These theories provide an essential analytical framework for China's pursuit of urban-rural integrated development.

Internationally, developed countries have cultivated relatively mature models for advancing urban-rural integration. The United States, for example, introduced the Smart Growth Program to establish metropolitan governance mechanisms, enhance infrastructure interconnectivity, and foster coordinated urban-rural planning. Germany implemented its "Urban-Rural Equalization" strategy, leveraging spatial planning laws and fiscal equalization to safeguard rural development rights. Japan successfully elevated rural industrial competitiveness through municipal mergers and the "One Village, One Product" initiative, while South Korea’s New Village Movement systematically upgraded rural production and living standards through government-guided, community-driven participation. These cases collectively underscore that land institutional reform, equitable public service allocation, and industrial synergy constitute pivotal pathways to achieving urban-rural integration. From the process of promoting urbanization and rural revitalization in China, it is evident that adjusting the irrational allocation of resources can achieve balanced, shared, and integrated development between urban and rural areas [1]. As a spatial carrier and production factor of urban and rural regional systems and production activities, land's use and allocation changes impact the circulation of resources between urban and rural areas, playing a crucial role in transforming the urban-rural structure and driving economic growth [2].Given that the allocation of land elements in China is primarily government-led, local governments not only provide the foundational conditions for regional economic development by selling industrial land at low prices to attract foreign investment and industrial agglomeration but also generate financial resources for regional economic development by selling commercial and residential land at high prices [3]. This aids in promoting urban-rural integration development. However, the long-standing urban-biased development approach in China has led to issues such as land segmentation, separation of people and land, and urban-rural division. These issues can result in irrational land sale practices that hinder urban-rural integration development [4].Thus, how do local government land sale practices impact urban-rural integration development? Addressing this question not only clarifies the mechanisms by which local government land sale practices influence urban-rural integration development but also provides theoretical and practical bases for local governments to use land resource allocation to promote urban-rural integration development. ”

For specific details, please refer to lines 55 to 92 of the manuscript.

3.The article is appropriately processed from a methodological point of view, but the limits of the research are missing in the conclusion. 

A Thank you very much for your careful review. Although the limitations of the study were discussed in the discussion section, based on your suggestion, this article has added the conclusion of the study and a summary of the limitations in the conclusion section. The specific summary is as follows:

“This study employs panel data from 281 prefecture-level and above cities in China, utilizing bidirectional fixed-effects models and moderated mediation models to analyze local governments' land conveyance practices. These practices are categorized into two dimensions—low-priced industrial land transfers and high-priced commercial/residential land transfers—and examined within a unified theoretical framework. The research systematically establishes the causal relationship between local governments' land conveyance strategies and urban-rural integrated development, while empirically analyzing their effects and underlying mechanisms. However, several limitations remain: (1) potential minor discrepancies between the constructed urban-rural integration index system and actual integration levels; (2) possible omission of other control variables that may influence urban-rural development; and (3) insufficient examination of whether moderating variables indirectly affect urban-rural integration through mediating variables.”

For specific details, please refer to lines 633 to 642 of the manuscript.

Once again, thank you very much for your comments and suggestions! Those comments are all valuable and very helpful for revising and improving our paper.

---

## [Decision Letter · Decision Letter 3]

Research on the Impact and Mechanism of Local Governments' Land Conveyance Behavior on Urban-Rural Integrated Development——Empirical Evidence from 281 Prefecture-Level Cities in China

PONE-D-24-33248R3

Dear Dr. Peng,

We’re pleased to inform you that your manuscript has been judged scientifically suitable for publication and will be formally accepted for publication once it meets all outstanding technical requirements.

Kind regards,

Amar Razzaq, PhD

Academic Editor

PLOS ONE

Additional Editor Comments (optional):

Reviewers' comments:

Reviewer's Responses to Questions

**Comments to the Author**

Reviewer #4: All comments have been addressed

2. Is the manuscript technically sound, and do the data support the conclusions?

Reviewer #4: Yes

3. Has the statistical analysis been performed appropriately and rigorously?

Reviewer #4: Yes

4. Have the authors made all data underlying the findings in their manuscript fully available?

Reviewer #4: Yes

5. Is the manuscript presented in an intelligible fashion and written in standard English?

Reviewer #4: Yes

Reviewer #4: Thank you for the detailed explanation of my comments and for incorporating them into the article. The edits contributed to improving the quality of the article.

**Do you want your identity to be public for this peer review?** For information about this choice, including consent withdrawal, please see our Privacy Policy

Reviewer #4: No

---

## [Editor Report · Acceptance letter]

PONE-D-24-33248R3

PLOS ONE

Dear Dr. Peng,

I'm pleased to inform you that your manuscript has been deemed suitable for publication in PLOS ONE. Congratulations! Your manuscript is now being handed over to our production team.

Kind regards,

on behalf of

Associate Professor Amar Razzaq

Academic Editor

PLOS ONE